# COLORBENCH: Can VLMs See and Understand the Colorful World? A Comprehensive Benchmark for Color Perception, Reasoning, and Robustness

**Yijun Liang,**\* **Ming Li,**\* **Chenrui Fan, Ziyue Li, Dang Nguyen, Kwesi Cobbina**
**Shweta Bhardwaj, Jiuhai Chen, Fuxiao Liu, Tianyi Zhou**
University of Maryland, College Park
{yliang17,minglii,tianyi}@umd.edu
Project: https://github.com/tianyi-lab/ColorBench

## Abstract

Color plays an important role in human perception and usually provides critical clues in visual reasoning. However, it is unclear whether and how vision-language models (VLMs) can perceive, understand, and leverage color as humans. This paper introduces "COLORBENCH", an innovative benchmark meticulously crafted to assess the capabilities of VLMs in color understanding, including color perception, reasoning, and robustness. By curating a suite of diverse test scenarios, with grounding in real applications, COLORBENCH evaluates how these models perceive colors, infer meanings from color-based cues, and maintain consistent performance under varying color transformations. Through an extensive evaluation of 32 VLMs with varying language models and vision encoders, our paper reveals some undiscovered findings: (i) The scaling law (larger models are better) still holds on COLORBENCH, while the language model plays a more important role than the vision encoder. (ii) However, the performance gaps across models are relatively small, indicating that color understanding has been largely neglected by existing VLMs. (iii) CoT reasoning improves color understanding accuracies and robustness, though they are vision-centric tasks. (iv) Color clues are indeed leveraged by VLMs on COLORBENCH but they can also mislead models in some tasks. These findings highlight the critical limitations of current VLMs and underscore the need to enhance color comprehension. Our COLORBENCH can serve as a foundational tool for advancing the study of human-level color understanding of multimodal AI.

## 1 Introduction

Color is widely recognized as a fundamental component of human visual perception [11, 34], playing a critical role and providing critical clues in object detection, scene interpretation, contextual understanding, planning, etc., across critical application scenarios such as scientific discovery, medical care, remote sensing, shopping, visualization, artwork interpretation, etc. For instance, [19] leverages spectral color signatures to distinguish vegetation, health, and water bodies in satellite imagery, and [1] utilizes sediment color patterns to detect marine ecosystems. These applications underscore how color-driven features play an important role in real-world scenarios. Moreover, colors can convey affective or semantic information beyond simply recognizing and naming colors since colors are highly correlated to other attributes or concepts and thus can provide key information to various downstream tasks that do not even directly ask about colors [18, 37, 45]. As modern vision-language models (VLMs) [12, 41, 48] continue to be deployed to increasingly diverse scenarios, color—an essential visual feature—plays a growing role in the processes of understanding and reasoning. It is

---

\*These authors contributed equally to this work.

39th Conference on Neural Information Processing Systems (NeurIPS 2025) Track on Datasets and Benchmarks.

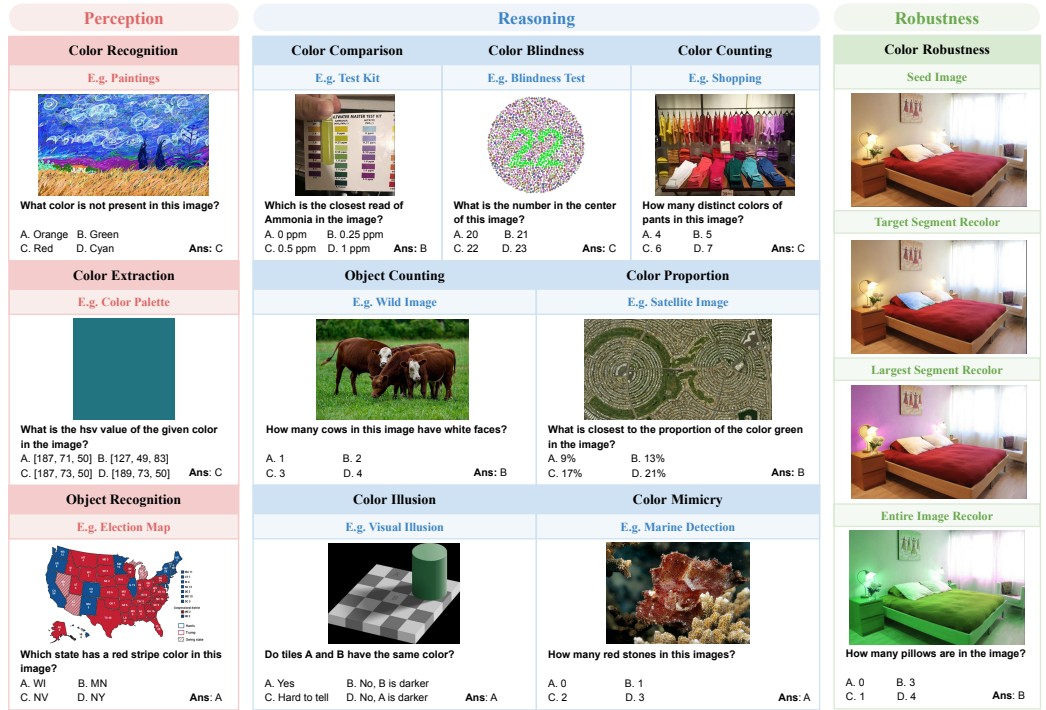

Figure 1: **Test samples from COLORBENCH.** COLORBENCH evaluates VLMs across three core capabilities: Perception, Reasoning and Robustness. The benchmark comprises 11 tasks designed to assess fine-grained color understanding abilities and the effect of color on other reasoning skills, including counting, proportion calculation, and robustness estimation. With over 1,400 instance, COLORBENCH covers a wide range of real-world application scenarios, including painting analysis, test kit readings, shopping, satellite/wildlife image analysis, etc.

essential to examine whether and how these models can understand and leverage color information as in human perception and reasoning, how color influences their overall perceptual and reasoning capabilities, and whether they can interpret visual illusions, resolve ambiguous cues, and maintain reliable performance under color variations.

However, existing benchmarks for VLMs mainly focus on tasks that may not heavily depend on color understanding or require color-centric reasoning, thereby overlooking nuanced color-related factors [25, 29]. Hence, there is a lack of benchmarks that systematically assess how well VLMs understand color when it serves as the main or distinguishing feature of a scene and key information to a task. Moreover, robustness to variations in color, such as recoloring and shifting hues, has also been largely neglected in the LLM era [6, 8, 20]. Consequently, **it remains unclear whether VLMs can perceive and reason about color with human-like proficiency and to what extent their performance deteriorates under significant color perturbations.** This shortfall underscores the need for a dedicated benchmark that comprehensively probes various facets of color comprehension in VLMs. A detailed discussion of related works is provided in Appendix A.

To bridge this gap, we propose a novel benchmark, **COLORBENCH**, that aims at comprehensively evaluating VLMs on three core capabilities of color understanding: **Color Perception**, **Color Reasoning**, and **Color Robustness**. Color Perception examines VLMs' fundamental capability to correctly detect and interpret colors from inputs. Color Reasoning refers to the reasoning skills to draw further conclusions based on the understanding of colors from input and prior knowledge, in which colors act as a crucial clue to formulate accurate judgments. Color Robustness assesses how consistently VLMs perform when an image's colors are altered, ensuring they maintain accurate predictions across different color variants of an image. Under these three core dimensions, 11 fine-grained tasks assessing different aspects of color understanding capabilities are formulated as shown in Figure 1, which not only shows test examples in COLORBENCH but also presents potential real-world applications.

By focusing on these facets, COLORBENCH offers a granular view of VLMs' capabilities in color understanding, aiming to illuminate both their strengths and shortcomings. We evaluate 32 widely

used VLMs in our benchmark, ranging from open-source to proprietary models, from relatively small models (0.5B) to larger models (78B), and obtain some unrevealed observations.

**Main Contribution.** We introduce "COLORBENCH", the first dedicated benchmark for assessing the color perception, reasoning, and robustness of VLMs. We develop an evaluation suite for 11 color-centric tasks, covering diverse application scenarios and practical challenges. Moreover, we report a fine-grained empirical evaluation of 32 state-of-the-art VLMs, which exposes their limitations in color understanding and offers novel insights for future research. Our key findings are highlighted in the following:

1. The scaling law still holds for color understanding but is much weaker and mainly depends on the language model parts. The correlation between the performance and the vision encoder's size is not significant due to the limited choices in current VLMs.

2. The absolute performances of different VLMs are relatively low, and the gaps between different models (open-source vs. proprietary, small vs. large) are not large, indicating the challenges of COLORBENCH and the negligence of color understanding in existing VLMs.

3. Despite the weaknesses of VLMs on color understanding, adding reasoning steps can still improve their performance on COLORBENCH tasks, even for color robustness, which has not been investigated by the community.

4. Color clues are indeed leveraged more or less by VLMs in most of the tasks in COLOR-BENCH. However, in color illusion and mimicry tasks, colors might mislead VLMs to give wrong answers, and converting colorful images into grayscale can improve the accuracy.

## 2 COLORBENCH Construction

We present **COLORBENCH**, the first benchmark explicitly designed to comprehensively evaluate the color understanding capabilities of VLMs across three key dimensions: **Color Perception**, **Color Reasoning**, and **Color Robustness**. This benchmark consists of $1,448$ instances and $5,814$ image-text questions spanning 11 diverse tasks. For the Color Perception and Color Reasoning categories, each instance contains an image, a question, and multiple-choice (3 to 6) options, with only one correct answer. For Color Robustness, each instance consists of 10 multiple-choice image-text questions, including a seed image and 9 edited images with color changes. Given that color is a fundamental visual feature influencing most vision-related tasks, disentangling color understanding from other general capabilities (e.g., object recognition, counting) is challenging. To address

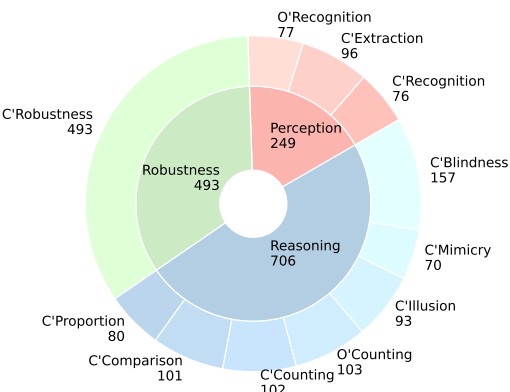

Figure 2: **Statistics** of 3 categories and 11 tasks in COLORBENCH.

this, we design questions with explicit color constraints for Color Perception and Reasoning dimensions, enabling a focused evaluation of VLMs' perception and reasoning abilities in relation to color.

### 2.1 Taxonomy

Motivated by the existing evaluation criteria from prior benchmarks and real-world application scenarios, we categorize the color understanding capability into 3 core dimensions and 11 detailed axes, as shown in Figure 1. The detailed question templates and sample cases are shown in Appendix D.

#### 2.1.1 Color Perception

This core dimension refers to the fundamental capability to correctly detect and interpret colors from inputs. We assess this capability through 3 key aspects: i) **Color Recognition**, ii) **Color Extraction**, and iii) **Object Recognition**.

**Color Recognition** includes questions that either ask for the color of a given object or determine whether a specific color is present in the image. **Color Extraction** requires the model to extract the

value of color code (e.g., RGB, HSV, or HEX) for a given single color image. This task measures the ability to perform fine-grained color retrieval from visual input. ***Object Recognition*** evaluates the model's capability to identify objects that match a specified color described in the text input. These two tasks require VLMs to be able to detect and interpret the color in either the image or text input.

### 2.1.2 Color Reasoning

This dimension refers to the reasoning skills to draw further conclusions based on the understanding of colors from input and prior knowledge, in which colors act as a crucial clue to formulate accurate judgments. This category encapsulates 7 key aspects: i) ***Color Proportion***, ii) ***Color Comparison***, iii) ***Color Counting***, iv) ***Object Counting***, v) ***Color Illusion***, vi) ***Color Mimicry*** and vii) ***Color Blindness***.

***Color Proportion*** tests the model's capability to estimate the relative area occupied by a specific color. Questions in this task require both color perception and proportion calculation capabilities. ***Color Comparison*** requires the model to be able to distinguish among multiple colors in the image, assessing its sensitivity to hue, saturation, and brightness differences in visual input. ***Color Counting*** focuses on identifying the number of unique colors in the image, evaluating the model's perception and differentiation of distinct color variations, and counting ability. ***Object Counting*** extends this challenge by requiring the model to count objects that match a specific color pattern. This task requires an integration of object recognition and color perception. ***Color Illusion*** questions query VLMs to compare colors in potential illusionary environments. This task evaluates the model's ability to account for color-induced optical illusions. ***Color Mimicry*** challenges the model to detect objects camouflaged within their surroundings, where color serves as a misleading factor, requiring advanced pattern recognition and contextual reasoning. These two tasks both assess the model's ability to make correct predictions under the misleading of color-related information in visual input. ***Color Blindness***, inspired by Ishihara tests, assesses the model's ability to recognize numbers or text embedded in color patterns, testing its understanding of shape-color relationships. These 7 tasks comprehensively assess the model's capacity for logical reasoning, spatial awareness, and adaptive interpretation of color-based visual cues.

### 2.1.3 Color Robustness

***Color Robustness*** assesses how consistently VLMs perform and whether they can consistently deliver accurate predictions under color variants of a given image. It involves measuring the stability of a VLM's responses when confronted with the same text input and a series of recolored images. To ensure that color does not influence the predictions, we select questions and corresponding answers that are independent of color attributes. Under these conditions, a robust model should produce unchanged predictions regardless of recoloring manipulation. Any variation in the model's responses is then used to quantify its susceptibility to color changes, providing a direct measure of robustness.

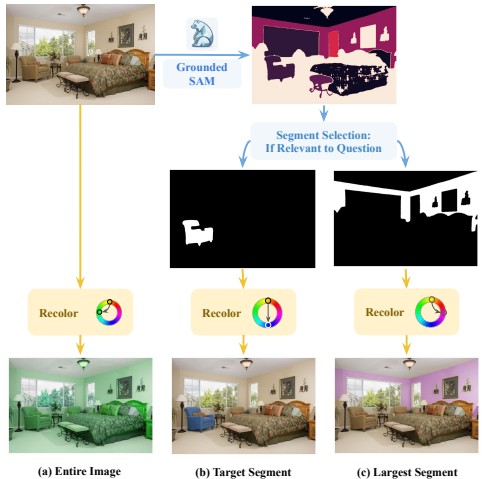

Figure 3: **Generation Pipeline for Color Robustness.** For each seed image, we apply 3 recoloring strategies (Entire Image, Target Segment, Largest Segment) to generate edited images. For each strategy, we change the color of the recoloring region via shifting the Hue values by $90°$, $180°$, or $270°$ in HSV color space.

### 2.2 Data Curation

For most of the tasks in the category of **Color Perception** and **Color Reasoning**, we rely on human experts to manually collect images from multiple online benchmarks and websites. For the Color Proportion task, to ensure the correctness of the ground truth, an extra color extraction tool is firstly utilized to obtain the color histogram of the image. Questions and options are then manually designed based on these color statistics. For tasks including Color Extraction, Color Blindness, and Color Illusion, testing images are generated by corresponding code programs to ensure the controllability of the questions and answers. The detailed data sources are shown in Appendix B.

After the initial data is collected, additional filtering processes are conducted in a human-machine interactive process. We first conduct inference on a variety of VLMs and discard low-quality samples based on the GPT-4o prediction result and human evaluation. For synthesized data, similar processes are conducted, but with additional code (for generation) and image assessment. The above process is conducted in three rounds before the final benchmark instances are settled. This refinement process ensures COLORBENCH a rigorous and informative benchmark for assessing color-related understanding.

For **Color Robustness**, we create evaluation instances by modifying images or specific regions through color changes. We define 3 recoloring strategies to determine the recoloring region: i) Entire Image, where the whole image is recolored; ii) Target Segment, where only the segment relevant to the question is altered; and iii) Largest Segment, where the largest region unrelated to the question is modified. Further details can be found in Appendix C. While generating color variants, we derive seed images from CV-Bench [42], a publicly available benchmark. For each seed image, as shown in Figure 3, we first employ a Grounded Segmentation Model (GAM) [38] to extract segments and their corresponding labels. We then apply the predefined recoloring strategies to determine the editing region and perform recoloring by shifting the Hue value in the HSV color space at three levels to cover entire color wheel: ($90°$, $180°$, and $270°$). This process produces 9 variations per seed image, covering different strategies and degrees of color change to enable a comprehensive robustness assessment. To ensure interpretability, human experts filter out unnatural or negligible modifications, resulting in a final selection of $493$ seed images for robustness evaluation.

## 2.3 Evaluation Metrics

For **Perception** and **Reasoning**, we use accuracy as the evaluation metric, as all tasks follow a multiple-choice format. Accuracy is computed per task and per category, representing the proportion of correctly answered questions.

For **Robustness**, we evaluate a model's ability to maintain consistent accurate predictions under color variations. As detailed in Section 2.2, each seed image $I_s$ is transformed into $n$ recolored variants using recoloring strategies, while keeping the original question $q$ unchanged. A model $\mathcal{M}$ is considered robust on a seed image $I_s$ and corresponding question $q$ if and only if it provides a correct prediction for $I_s$ and maintains correct on all $n$ recolored versions. To quantify robustness, we define the instance-level robustness metric $R(I_s, q) \in \{0, 1\}$ and a model-level robustness metric $Robust_{\mathcal{M}} \in [0, 1]$.

**Instance-level Robustness.** Let the recolored images be $I_1, \cdots, I_n$ and the generation output of model for image $I_i$ and question $q$ is $\mathcal{M}(I_i, q)$. Define $c(\mathcal{M}(I_i, q))$ as the model correctness: $c(\mathcal{M}(I_i, q)) = 1$ if model result $\mathcal{M}(I_i, q)$ is correct, otherwise $0$. The instance-level robustness metric $R(I_s, q)$ for a seed image $I_s$ and question $q$ is defined as:

$$R(I_s, q) = \begin{cases} 1 & \text{if } c(\mathcal{M}(I_i, q)) = c(\mathcal{M}(I_s, q)) = 1, \forall i \in [n] \\ 0 & \text{otherwise} \end{cases} \tag{1}$$

**Overall Robustness.** Let $\mathcal{S}$ be the set of seed images. We define model robustness to be:

$$Robust_{\mathcal{M}} = \frac{\sum_{I_s \in \mathcal{S}} R(I_s)}{|\mathcal{S}|}, Robust_{\mathcal{M}} \in [0, 1] \tag{2}$$

$Robust_{\mathcal{M}}$ represents the proportion of seed images on which the model maintains correctness across all color variations. A model is more robust when $Robust_{\mathcal{M}}$ is higher.

# 3 Experimental Results

## 3.1 Main Results

Table 1 presents the performances of a wide range of VLMs, along with human evaluation results on our COLORBENCH. Human participants achieve the highest performance on all evaluated tasks across all models. Among the models, overall accuracy generally increases with model size, with larger models tend to outperform smaller models, and the two proprietary models, GPT-4o and Gemini-2-flash, perform the best[2].

---

[2]To examine the upper limits of VLM capabilities and benchmark against human-level performance, we also assess performance GPT-o3 on perception and reasoning tasks. The result is shown in Appendix H.

Table 1: **Performance of** 32 **VLMs (grouped by size) and human performance on COLORBENCH.** Models are ranked within each group according to their overall performance on Color Perception and Reasoning (P & R Overall) tasks. For human evaluation, Color Extraction task is excluded, as humans are not attuned to precise color code differences. The best performance in each VLM group is highlighted in bold. For human evaluation, any instance surpassing all VLMs is marked in bold.

| | Color Perception | | | Color Reasoning | | | | | | | P & R | Robustness |
|---|---|---|---|---|---|---|---|---|---|---|---|---|
| | C'Recog | C'Extract | O'Recog | C'Prop | C'Comp | C'Count | O'Count | C'Illu | C'Mimic | C'Blind | Overall | C'Robust |
| *VLMs: < 7B* | | | | | | | | | | | | |
| LLaVA-OV-0.5B | 26.3 | **44.8** | 46.8 | 30.0 | 23.8 | 22.6 | 21.4 | 38.7 | 58.6 | 26.8 | 32.6 | 38.7 |
| InternVL2-1B | 35.5 | 34.4 | 59.7 | 23.8 | 41.6 | 19.6 | 22.3 | 34.4 | 38.6 | **33.1** | 33.6 | 39.4 |
| InternVL2-2B | 60.5 | 36.5 | 66.2 | 40.0 | 38.6 | 19.6 | 29.1 | 26.9 | 52.9 | 21.0 | 36.4 | 54.2 |
| InternVL2.5-1B | 55.3 | 36.5 | 61.0 | 42.5 | 45.5 | 22.6 | 25.2 | 43.0 | 41.4 | 28.0 | 38.3 | 52.3 |
| InternVL2.5-2B | 69.7 | 28.1 | 71.4 | 33.8 | 48.5 | **25.5** | **30.1** | 32.3 | 55.7 | 19.8 | 38.5 | 59.8 |
| Qwen2.5-VL-3B | **72.4** | 38.5 | **74.0** | 43.8 | 48.5 | 22.6 | 25.2 | 43.0 | 45.7 | 24.2 | 41.1 | **63.7** |
| Cambrian-3B | 67.1 | 31.3 | 66.2 | **47.5** | **50.5** | **25.5** | 29.1 | **44.1** | **61.4** | 22.3 | **41.5** | 59.0 |
| *VLMs: 7B − 8B* | | | | | | | | | | | | |
| LLaVA-Next-v-7B | 29.0 | 38.5 | 57.1 | 21.3 | 34.7 | 23.5 | 25.2 | 38.7 | 41.4 | 17.8 | 31.2 | 52.1 |
| LLaVA-Next-m-7B | 21.1 | 18.8 | 63.6 | 27.5 | 42.6 | 16.7 | 34.0 | 41.9 | 47.1 | **29.9** | 33.4 | 55.2 |
| Eagle-X5-7B | 52.6 | 47.9 | 67.5 | 41.3 | 42.6 | 20.6 | 35.0 | 44.1 | 48.6 | 22.9 | 40.0 | 48.5 |
| Cambrian-8B | 72.4 | 28.1 | 72.7 | 48.8 | 54.5 | **31.4** | 33.0 | 41.9 | **57.1** | 17.2 | 42.3 | 64.9 |
| InternVL2-8B | 72.4 | 50.0 | 77.9 | 42.5 | 48.5 | 20.6 | 35.9 | 38.7 | 50.0 | 23.6 | 43.1 | 65.5 |
| Eagle-X4-8B | 71.1 | 47.9 | 68.8 | 45.0 | 50.5 | 26.5 | **37.9** | 40.9 | 48.6 | 27.4 | 44.1 | 63.7 |
| LLaVA-OV-7B | 71.1 | **53.1** | 81.8 | 52.5 | 53.5 | 19.6 | 26.2 | **48.4** | 48.6 | 23.6 | 44.7 | 74.0 |
| InternVL2.5-8B | **77.6** | 47.9 | 83.1 | **50.0** | **62.4** | 25.5 | 33.0 | 34.4 | 52.9 | 19.8 | 45.2 | 69.8 |
| Qwen2.5-VL-7B | 76.3 | 49.0 | **84.4** | 47.5 | 52.5 | 19.6 | 34.0 | 44.1 | 55.7 | 28.7 | **46.2** | **74.4** |
| *VLMs: 10B − 30B* | | | | | | | | | | | | |
| LLaVA-Next-13B | 56.6 | 31.3 | 71.4 | 27.5 | 41.6 | 27.5 | 28.2 | 29.0 | 45.7 | 25.5 | 36.4 | 53.3 |
| Cambrian-13B | 67.1 | 34.4 | 74.0 | 46.3 | 47.5 | **32.4** | 35.0 | 38.7 | 55.7 | 24.8 | 42.8 | 64.7 |
| Eagle-X4-13B | **73.7** | 43.8 | 76.6 | 43.8 | 47.5 | 23.5 | **38.8** | 34.4 | **57.1** | 26.1 | 43.7 | 66.3 |
| InternVL2-26B | 72.4 | **52.1** | 87.0 | **52.5** | 56.4 | 20.6 | 35.0 | 34.4 | 55.7 | 27.4 | 46.3 | 74.0 |
| InternVL2.5-26B | 72.4 | 45.8 | **89.6** | 45.0 | **63.4** | 22.6 | 35.0 | 32.3 | 62.9 | **29.3** | **46.8** | **83.0** |
| *VLMs: 30B − 70B* | | | | | | | | | | | | |
| Eagle-X5-34B | **79.0** | 27.1 | 80.5 | 48.8 | 48.5 | 23.5 | 35.9 | **37.6** | 60.0 | 25.5 | 43.4 | 67.1 |
| Cambrian-34b | 75.0 | 57.3 | 77.9 | 50.0 | 46.5 | 22.6 | 32.0 | **37.6** | **64.3** | 24.2 | 45.3 | 67.7 |
| InternVL2-40B | 72.4 | 52.1 | 83.1 | 51.3 | 61.4 | 19.6 | 35.9 | 34.4 | 58.6 | 21.0 | 45.6 | 78.7 |
| LLaVA-Next-34b | 69.7 | 46.9 | 76.6 | 43.8 | 56.4 | 28.4 | **41.8** | 36.6 | 61.4 | **29.9** | 46.6 | 65.9 |
| InternVL2.5-38B | 71.1 | **60.4** | **89.6** | **53.8** | **63.4** | **29.4** | 40.8 | 34.4 | 61.4 | 26.8 | **50.0** | **84.6** |
| *VLMs: > 70B* | | | | | | | | | | | | |
| InternVL2-76B | 72.4 | 42.7 | **85.7** | 45.0 | 62.4 | **27.5** | 35.0 | 31.2 | 50.0 | 23.6 | 44.6 | 68.6 |
| LLaVA-Next-72B | 72.4 | 54.2 | 79.2 | 41.3 | 49.5 | 24.5 | 35.9 | 33.3 | 48.6 | **34.4** | 45.2 | 66.5 |
| InternVL2.5-78B | **75.0** | 58.3 | 81.8 | 43.8 | 68.3 | 27.5 | 36.9 | 34.4 | 61.4 | 28.7 | 48.8 | **86.2** |
| LLaVA-OV-72B | 73.7 | **63.5** | 83.1 | **52.5** | **69.3** | 27.5 | **50.5** | **36.6** | 55.7 | 31.9 | **51.9** | 80.3 |
| *VLMs: Proprietary* | | | | | | | | | | | | |
| GPT-4o | 76.3 | 40.6 | 80.5 | 38.3 | 66.3 | 30.4 | 29.1 | **50.5** | 70.0 | 58.6 | 52.9 | 46.2 |
| Gemini-2-flash | 80.3 | 52.1 | 87.0 | 46.9 | 70.3 | 33.3 | 34.9 | 44.1 | 72.9 | 49.6 | 55.4 | 70.7 |
| GPT-4o (CoT) | 77.6 | 55.2 | 83.1 | 44.4 | **71.3** | 26.5 | 33.0 | 44.1 | **77.1** | 66.8 | 57.4 | 69.9 |
| Gemini-2-flash (CoT) | **82.9** | 56.2 | **88.3** | **58.0** | 68.3 | **43.1** | 38.8 | 40.9 | 75.7 | 60.0 | **59.6** | 73.6 |
| *Human Evaluation* | | | | | | | | | | | | |
| Human Evaluation | **92.0** | - | **90.1** | 59.6 | 79.8 | 62.0 | 81.3 | 63.0 | 83.8 | 94.0 | - | - |

**Color Perception.** In *Color Recognition (C'Recog)*, most models perform well (above $60\%$), indicating that this task is relatively basic for color perception. Gemini-2 with CoT obtains the highest performance. In *Color Extraction (C'Extra)*, to our surprise, the two powerful proprietary models without CoT prompting only reach the middle-tier performances, indicating the potential limitation on the color perception of their vision encoders. Similar to the Color Existence task, almost all the models perform well in *Object Recognition (O'Recog)*, and the 2 proprietary models do not reach the top. This is probably due to the strong alignment between this task and the common training recipe, which includes abundant general object detection images.

**Color Reasoning.** In *Color Proportion (C'Prop)*, even the best model, Gemini-2 with CoT, can only reach $58.0\%$ of the accuracy, which is almost only slightly better than random guessing, showcasing the supreme difficulty of this task. In *Color Comparison (C'Comp)*, larger models perform better in this task, and the proprietary models with CoT reach the top performance unsurprisingly. Surprisingly, in *Color Counting (C'Count)*, all models show extremely poor performances. The highest performance comes from Gemini-2 with CoT, exceeding the second place by 10 percent, although its performance is also unsatisfactory at only $43.1\%$. In *Object Counting (O'Count)*, surpassing the 2 proprietary models, LLaVA-OV-72B reaches the top and becomes the only model that exceeds $50\%$ of the accuracy. Similar to the findings from the Object Recognition task, this might be caused by the extremely adequate object detection tasks in open-sourced training recipes. In *Color Illusion (C'Illu)*, the accuracies of most models lie in the range of $30\%$ to $50\%$, and GPT-4o without CoT is the only one that exceeds $50\%$ of the accuracy. In *Color Mimicry (C'Mimic)*, the 2 proprietary models reach the top, while more reasoning steps do not benefit a lot. In *Color Blindness (C'Blind)*, most of the open-sourced models present accuracies under $30\%$. Considering the extremely practical usage of this scenario, we think the current community should pay more attention to this. Moreover,

Table 2: Spearman's rank correlation between VLM performance and different model parts' sizes on each task. **L** denotes the language model part's size and **V** represents the vision encoder part's size. We use "(*)" to mark correlations with p-values $\leq 0.05$ **It shows that the scaling law still holds for color understanding but it is much weaker.**

|  | Color Perception | | | Color Reasoning | | | | | | | P & R | Color Robustness |
|---|---|---|---|---|---|---|---|---|---|---|---|---|
|  | C'Recog | C'Extract | O'Recog | C'Prop | C'Comp | C'Count | O'Count | C'Illu | C'Mimic | C'Blind | Overall | C'Robust |
| **L+V** | 0.5657 (*) | 0.5255 (*) | 0.7107 (*) | 0.5125 (*) | 0.6358 (*) | 0.4316 (*) | 0.7566 (*) | -0.3460 | 0.4832 (*) | 0.2460 | 0.7619 (*) | 0.7386 (*) |
| **L** | 0.5724 (*) | 0.4937 (*) | 0.6769 (*) | 0.4696 (*) | 0.6118 (*) | 0.4408 (*) | 0.7611 (*) | -0.3697 (*) | 0.4559 (*) | 0.2824 | 0.7436 (*) | 0.7123 (*) |
| **V** | 0.3955 (*) | 0.2856 | 0.5465 (*) | 0.6242 (*) | 0.5295 (*) | 0.2089 | 0.3608 | -0.0127 | 0.6024 (*) | -0.0679 | 0.5271 (*) | 0.5623 (*) |

we also observe that, surprisingly, more reasoning steps benefit VLMs in the color blindness test, although it seems like a pure color perception task.

**Color Robustness.** In *Color Robustness (C'Robust)*, a higher value represents better robustness towards color alteration. The only 4 models that exceed 80% are LLaVA-OV-72B, InternVL2.5-26B, InternVL2.5-38B, and InternVL2.5-78B, which utilize relatively larger vision encoders, InternViT-6B, compared with others (mostly only 300-400M). In the meantime, GPT-4o has a really low robustness (46.2%) to colors, indicating its vulnerable sensitivity to color changes, while Gemini-2 shows promising robustness (70.7%) towards colors. Moreover, another surprising observation is that even though only the colors are changed and all the original queries are kept, utilizing more reasoning steps can consistently improve robustness for GPT-4o (+23.7%) and Gemini-2 (+2.9%).

## 3.2 Further Findings

> **Finding 1.** The scaling law still holds for color understanding, but is much weaker and mainly depends on the language model parts. The correlation between the performance and the vision encoder's size is not significant due to the limited choices in current VLMs.

Since color-related tasks often involve abstract reasoning, language comprehension, and contextual interpretation, it is essential to assess not just the vision encoder but also part of the language model, which plays a critical role in processing and understanding such tasks. To quantitatively analyze the correlation between VLM performances on color understanding tasks and their sizes, Spearman's rank correlation is calculated between VLM performances and (i) overall model sizes (**L** + **V**), (ii) language model sizes (**L**), and (iii) vision encoder sizes (**V**). The correlation values and p-signs are presented in Table 2; a star is notated when the p-value of the correlation is lower than 0.05. It is observed that between the performances and language model

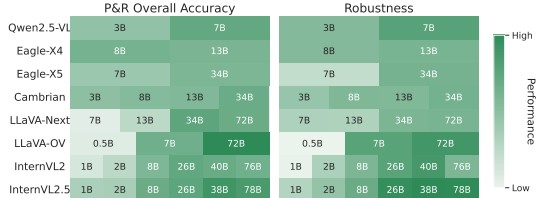

Figure 4: **The heatmaps related to performances and VLM sizes.** Deeper color represents higher performance of P&R Overall Accuracy or Robustness. Each line represents a model family with the sizes growing from small to large. This visualization clearly shows the correlation between performances and model sizes, larger model leads to higher performance.

sizes, most of the tasks have a correlation greater than 0.5 and a p-value smaller than 0.05, except for Color Illusion and Color Blindness due to their special characteristics. Since the correlation between overall model sizes (**L** + **V**) and P&R Overall (0.7619), and Robustness (0.7390), we conclude that the color understanding, including Color Perception, Color Reasoning, and Color Robustness, still follows the scaling law of model sizes. Figure 4 presents the correlations between performances and model sizes in each model family. This visualization clearly shows the correlation between performances and model sizes; a larger model leads to higher performance within each model family.

However, between the performances and vision encoder sizes, most of the tasks either have a correlation lower than 0.5 or a p-value greater than 0.05, which is not sufficient to conclude with the evident positive correlation. Despite these findings, we try to avoid conveying the message that there is no positive correlation between performances and vision encoder sizes. We think it is because of the negligence of the current community to focus on the scaling laws of vision encoders. The vision encoders used in the current mainstream VLMs are constrained in a very small set: (i) most of the VLMs only use one type of vision encoders for the whole family, except for the InternVL2 and InternVL2.5 series; (ii) most of the VLMs use the vision encoder with the size of 300 - 400M. These challenges make it hard to evaluate the scaling laws of vision encoders. Further visualizations are presented in Appendix L.2.

Table 4: **Adding reasoning steps can improve VLMs' performance on COLORBENCH.** The change of accuracy brought by Chain of Thought (CoT) prompting on all tasks for GPT-4o and Gemini-2-flash. The last row presents the average improvement across both models.

| | Color Perception | | | Color Reasoning | | | | | | | P & R | Color Robustness |
|---|---|---|---|---|---|---|---|---|---|---|---|---|
| | C'Recog | C'Extract | O'Recog | C'Prop | C'Comp | C'Count | O'Count | C'Illu | C'Mimic | C'Blind | Overall | C'Robust |
| GPT-4o △ | +1.3 | +14.6 | +2.6 | +6.1 | +5.0 | −3.9 | +3.9 | −6.4 | +7.1 | +8.2 | +4.5 | +23.7 |
| Gemini-2 △ | +2.6 | +4.1 | +1.3 | +11.1 | −2.0 | +9.8 | +3.9 | −3.2 | +2.8 | +10.4 | +4.2 | +2.9 |
| Average △ | +1.95 | +9.35 | +1.95 | +8.60 | +1.50 | +2.95 | +3.9 | −4.80 | +4.95 | +9.30 | +4.35 | +13.30 |

> **Finding 2.** The absolute performances of different VLMs are relatively low and lag behind those of humans. Moreover, the gaps between different models (open-source vs. proprietary, small vs. large) are not large, indicating the challenges of COLORBENCH and the negligence of color understanding in existing VLMs.

As shown in Table 3, we separate all the VLMs into several groups based on their sizes and present the best accuracy and the model name within each group. We can see that even the powerful proprietary models, GPT-4o and Gemini-2, can only reach an overall color perception and reasoning (P & R Overall) accuracy of 53.9%, only +2.0% better than the best open-sourced model. Task-level results in Table 1 further reveal that these advanced proprietary models still exhibit substantial performance gaps compared to humans across most tasks. Moreover, the best model from group 1 has the accuracy of 41.5% (Cambrian-3B), which is only 10.4% lower than the best open-sourced

Table 3: The best model within each group and its performances (on P&R accuracy and Robustness). **The absolute performances of different VLMs on COLORBENCH are relatively low, and the performance gaps between models are not large.**

| | Color P & R Overall | | Color Robustness | |
|---|---|---|---|---|
| Model Size | Model | Best | Model | Best |
| <7B | Cambrian-3B | 41.5 | Qwen2.5-VL-3B | 63.7 |
| 7B–8B | Qwen2.5-VL-7B | 46.2 | Qwen2.5-VL-7B | 74.4 |
| 10B–30B | InternVL2.5-26B | 46.8 | InternVL2.5-26B | 83.0 |
| 30B–50B | InternVL2.5-38B | 50.0 | InternVL2.5-38B | 84.6 |
| >70B | LLava-OV-72B | 51.9 | InternVL2.5-78B | 86.2 |
| Proprietary | Gemini-2 | 55.4 | Gemini-2 | 70.7 |
| Proprietary | Gemini-2 (CoT) | 59.6 | Gemini-2 (CoT) | 73.6 |

model. As for the robustness, the powerful proprietary models even show weaker robustness than the 7B model. Considering the lack of existing benchmarks specifically evaluating VLMs' color understanding capabilities, we conclude that this area is long-neglected by the community, and the open-sourced community is still on the same page with the proprietary model providers.

> **Finding 3.** Despite the weaknesses of VLMs on color understanding, adding reasoning steps can still improve their performance on COLORBENCH tasks, even for color robustness, which has not been investigated by the community.

The impact of using CoT prompting is shown in Table 4, in which we can see CoT improves the average P&R Overall accuracy across both models by +4.35%, indicating that reasoning benefits these color-related tasks. Within the category of Color Perception, the improvements from CoT on Color Recognition and Object Recognition are quite limited as these tasks heavily rely on the vision encoder. Figure 59 and 60 in Appendix M illustrate that adding reasoning steps does not take effect since the initial visual perception and color identification are incorrect in the slow thinking process. However, to our surprise, we find that the Color Extraction task benefits extremely from more reasoning steps, although it seems only related to the vision encoder. After a thorough investigation, we observe that most of the current VLMs are not capable of directly extracting color values, so they need to use more reasoning steps to reach reasonable answers.

Within the category of Color Reasoning, CoT benefits most of the tasks. However, in the Color Illusion task, CoT harms the model performance. After a manual investigation, we observe that more reasoning steps might cause VLMs to focus more on the misleading environments rather than directly compare the assigned colors, as shown in Figure 61. Another observation occurs in the Color Blindness task. Unlike other reasoning-related tasks, humans can read a color blindness test image with a simple glimpse without any slow thinking. This fascinating misalignment between humans and VLMs intrigues us to further investigation. We find that VLMs recognize these digits in a button-up pattern: they need to first infer that the dots in the image can form a digit before they really recognize these dots as digits.

In addition, the consistent improvement of CoT on Color Robustness is also an unrevealed phenomenon. In our setting, only the colors of the image are altered, and the questions are strictly the

same as the original. Thus, under this circumstance, color is the only variant, which is supposed to be more related to the capability of the vision encoder. However, counterintuitively, as shown in our experiments, more reasoning steps make the VLMs more robust to the color changes, which is probably caused by the higher confidence of correct answers after reasoning.

> **Finding 4.** Color clues are indeed leveraged more or less by VLMs in most of the tasks in COLORBENCH. However, in color illusion and mimicry tasks, colors might mislead VLMs to wrong answers, and converting colorful images to grayscale can improve the accuracy.

In order to examine whether VLMs really leverage color clues to handle tasks in COLORBENCH, experiments are conducted by converting all the original colorful images in the Color Perception and Reasoning categories into gray-scale ones, without changing the questions. Under this circumstance, the accuracies are expected to decrease dramatically as all our questions are related to colors. For quantitative analysis, we calculate the accuracy changing ratio as $(Acc_{ori} - Acc_{gray})/Acc_{ori}$ for each VLM on each task. This value directly represents how the original accuracy changes with a gray-scale transformation. The positive value represents that the VLM has a higher accuracy on the original colored images, indicating that it needs color clues to solve the task. Higher positive values represent higher significance of the color clues. On the contrary, if the value is negative, it means that the VLM can reach a better accuracy after the gray-scale transformation, indicating that it does not need

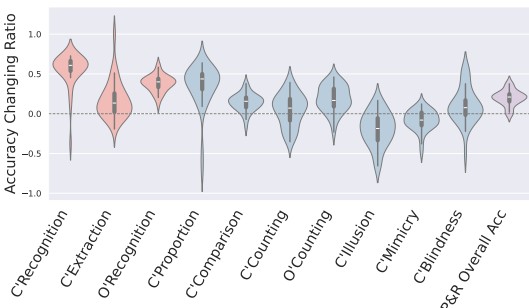

Figure 5: The percentage of change in accuracy (y-axis) by converting colorful images to grayscale in each COLORBENCH task (x-axis). Each violin plot visualizes the distribution over all VLMs. Higher (lower) percentage indicates that VLMs rely more (less) on color clues for the task. Positive (negative) percentage indicates degradation (improvement) on grayscale images. **Color clues are indeed more or less leveraged by VLMs in most tasks but they might mislead VLMs (illusion & mimicry).**

color clues for the task, and colors might even mislead VLM's judgment. Lower negative values represent the severe harm the color can have on the task.

The accuracy changing ratio distributions across all VLMs and tasks are presented in Figure 5 as the violin plot. As shown in the figure, for most of the tasks, the ratios of VLMs are above 0, indicating that VLMs indeed leverage color clues to correctly solve the tasks; removing the color directly harms the original accuracies dramatically. However, when it comes to Color Illusion and Color Mimicry, the majority of the changing ratios are below 0, which means that VLMs can get better accuracies when all the color information is removed. This phenomenon is reasonable as the colors on both of these two tasks are more likely serving as the misleading factors. In the meantime, for the Color Counting and Color Blindness tasks, almost half the accuracies increase and half decrease, indicating that the color clues might not be so significant in this task, thus, some of the models can find other ways to get the answer. We also investigate the correlation between accuracy changing ratios and model sizes, while no significant correlation can be concluded.

## 4  Conclusion, Limitation, and Future Works

In this paper, we introduce COLORBENCH, the first benchmark designed to comprehensively evaluate the color understanding capabilities of VLMs, including Perception, Reasoning, and Robustness. After evaluating 32 widely used VLMs on our benchmark, several undiscovered observations are revealed by us. These observations emphasize the need for more sophisticated model architectures that integrate deeper color reasoning capabilities. To ensure high-quality and reliable annotations, COLORBENCH relies on manual data collection, annotation, and assessment across most domains. While this guarantees consistency, it inevitably limits dataset scale, style diversity, and category coverage. As future work, we aim to develop a trustworthy automated data collection pipeline and expand COLORBENCH to larger-scale, more diverse tasks involving complex interplays of color with texture, shape, and spatial relationships. Furthermore, investigating the impact of different visual encoders and language models could further elucidate the pathways through which VLMs process color information.

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

# Table of Contents for Appendix

# A Related Works

## A.1 VLM Benchmarks

With the rapid advancements in Vision-Language Models (VLMs) [9], numerous benchmarks have emerged to systematically evaluate VLM capabilities across diverse dimensions [29]. These benchmarks generally fall into two categories: text-centric and vision-centric evaluations, each designed to assess distinct multimodal competencies. Text-centric benchmarks primarily measure commonsense knowledge, reasoning, and complex problem-solving capabilities, exemplified by tasks in MMMU [47] and NaturalBench [23]. Conversely, vision-centric benchmarks focus on visual perception and reasoning (MMBench [32] and MME [10]), and robustness to visual perturbations (Grit [14] and Visual Robustness [17]). Furthermore, several benchmarks have extended their scope to evaluate specialized visual tasks, such as spatial relationship comprehension (SEED-Bench [22] and MM-Vet [46]), chart and map understanding (MMSTAR [4] and MuirBench [43]), visual grounding (Flickr30k [36] and TRIG [27]) and the detection and understanding of visual hallucinations (POPE [28] and HallusionBench [13]). However, despite the extensive scope covered by existing VLM benchmarks, none currently provide an integrated evaluation that simultaneously assesses visual perception, reasoning, and robustness within a unified framework. Moreover, although certain benchmarks [32, 10] have incorporated color-related questions, these have typically addressed basic color perception and recognition, neglecting deeper assessments of reasoning and robustness associated with color understanding.

## A.2 Color Evaluation

Color understanding is increasingly recognized as a crucial aspect of Vision-Language Models' ability to perceive and interpret visual content. Limited studies have explored how color information influences model performance on specific tasks. Some studies [51, 50] explore the understanding of color by replacing color-related words in textual inputs to evaluate the models' ability to handle color-specific information. More recent research [16, 21] focuses on assessing fine-grained color discrimination by asking models to distinguish subtle color differences in visual inputs. Samin et al. [39] introduced color-related foils to test VLMs' capacity to cognize basic colors like red, white, and green, particularly in contexts requiring attention to subtle cues. Additionally, Burapacheep et al. [3] developed a benchmark dataset to evaluate and enhance compositional color comprehension in VLMs, emphasizing tasks where understanding minimal color relationships is essential. IllusionVQA [40] evaluates model perception of color illusions in photorealistic scenes. While these works have addressed isolated aspects of color understanding, none have provided a holistic assessment framework. In contrast to these previous works, our study establishes the first comprehensive and specialized benchmark for evaluating the color-related abilities of VLMs, offering a quantitative, automated approach to further this area of research.

# B Data Sources

We conduct COLORBENCH from multiple sources, including website sources, publicly available benchmarks, and generated images. The detailed sources are included in Table 5.

Table 5: **Data sources for each task.**

| Category | Data Source |
|---|---|
| C'Recognition | Website, ICAA17K [15] |
| C'Recognition | Website, ICAA17K [15] |
| C'Extraction | Synthetic Data |
| C'Proportion | Website, Synthetic Data |
| C'Comparison | Website |
| C'Counting | Website, Synthetic Data |
| C'Ounting | Website, ADA20K [52, 53], COCO2017 [30] |
| C'Mimicry | Website, IllusionVQA[40], RCID[33] |
| C'Blindness | Synthetic Data |
| C'Robust | CV-Bench[42] |

Table 6: **Recoloring strategies.**

| Strategy | Editing Region | Purpose |
|---|---|---|
| Entire Image | Whole image | Assesses the model's robustness to global color shifts |
| Target Segment | Segment containing the object referenced in the question | Evaluates the model's sensitivity to task-relevant color changes |
| Largest Segment | The largest segment that is irrelevant to the question | Tests whether changes in dominant but unrelated regions affect model predictions |

## C  Detailed Generation Process for Robustness

For the **Color Robustness**, we evaluate the consistency of VLMs when faced with instances that differ only in the color of the visual input. To systematically assess this effect, we define 3 recoloring strategies that determine which part of the image is altered: i) **Target Segment**, ii) **Largest Segment**, and iii) **Entire Image**. As mentioned in Table 6, **Target Segment** strategy recolors only the segment containing the object referenced in the question. This strategy ensures that the modification directly affects the model's perception of task-relevant content. **Largest Segment** strategy alters the color of the largest segment that is irrelevant to the question, testing whether models are distracted by dominant but unrelated visual changes. In contrast, **Entire Image** strategy applies a global color shift to evaluate the model's sensitivity to overall color variations. As summarized in Table 6, the first two strategies introduce localized modifications, while the third assesses robustness to broader image-wide color changes. Importantly, only color attributes are altered without modifying object shapes or contextual elements, which preserves the overall realism of the image. By incorporating both task-relevant and irrelevant edits, our benchmark provides a comprehensive evaluation of VLMs' ability to handle color perturbations across different contexts.

While generating color variations, we derive seed images from CV-Bench [42], a publicly available benchmark. For each seed image, as shown in Figure 3, we first employ a Grounded Segmentation Model (GAM) [38] to extract segments and their corresponding labels. We then apply the predefined recoloring strategies to determine the editing region. Once the editing region is determined, we modify the color of the corresponding region. In HSV color space, since Saturation and Value control the purity or brightness of the color, and only Hue controls the color of the part, we only adjust the Hue value in the HSV color space. Specifically, we shift the Hue by 90°, 180°, and 270°. These three values ensure that the color manipulations cover significant perceptual differences across the color spectrum. This process produces nine variations per seed image, covering different strategies and degrees of color change to enable a comprehensive robustness assessment. To ensure interpretability, human experts filter out unnatural or negligible modifications, resulting in a final selection of 493 seed images for robustness evaluation. Additionally, we select questions that are color-invariant, which means answers remain valid regardless of whether the recoloring appears fully natural. This design choice isolates color variation as the sole variable of interest and prevents confounding effects from semantic or contextual changes. Through these steps, we evaluate whether VLMs rely excessively on color information and whether they maintain consistency in their predictions despite substantial color shifts.

## D  COLORBENCH Categories and Questions

Table 7 provides a detailed description of each task, alongside representative figures and sample questions that effectively demonstrate the specific capabilities being tested. Cases are provided for each task in Figure 6 to 16.

Table 7: **Task and question definition in COLORBENCH.**

| | Task | # | Sample Case | Description | Sample Questions |
|---|---|---|---|---|---|
| Perception | Color Recognition | 76 | Figure 6 | Ask for the color of a specific object or determine if a particular color is present in the image. | What is the color of *object* in this image? What color does not exist in this image? |
| | Color Extraction | 96 | Figure 7 | Extract the color code value (e.g., RGB, HSV, or HEX) from a single color in the image. | What is the HSV value of the given color in the image? What is the RGB value of the given color in the image? |
| | Object Recognition | 77 | Figure 8 | Identify objects in the image that match a specified color noted in the text input. | What *object* has a color of *pink* in this image? |
| Reasoning | Color Proportion | 80 | Figure 9 | Estimate the relative area occupied by a specified color in the image. | What is the dominant color in this image? What is the closest to the proportion of the red color in the image? |
| | Color Comparison | 101 | Figure 10 | Distinguish among multiple colors present in the image to assess overall tones and shades. | Which photo is *warmer* in overall color? Which object has a *darker* color in the image? |
| | Color Counting | 102 | Figure 11 | Identify the number of unique colors present in the image. | How many different colors are in this image? |
| | Object Counting | 103 | Figure 12 | Count the number of objects of a specified color present in the image. | How many objects with *green* color are in this image? |
| | Color Illusion | 93 | Figure 13 | Assess and compare colors in potential illusionary settings within the image. | Do two objects have the same color? |
| | Color Mimicry | 70 | Figure 14 | Detect objects that are camouflaged within their surroundings, where color is a key deceptive element. | How many *animals* are in this image? |
| | Color Blindness | 157 | Figure 15 | Recognize numbers or text that are embedded in color patterns, often used in tests for color vision. | What is the number in the center of the image? |

## Color Recognition

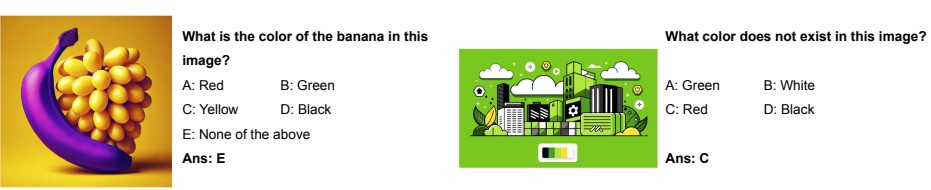

Figure 6: **Cases for Color Recognition Task.**

## Color Extraction

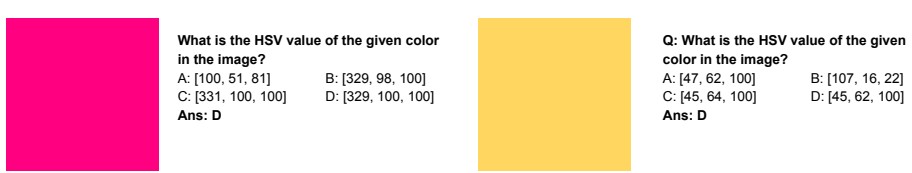

Figure 7: **Cases for Color Extraction Task.**

**Object Recognition**

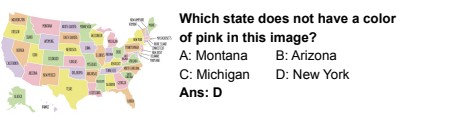
Which state does not have a color of pink in this image?
A: Montana    B: Arizona
C: Michigan    D: New York
**Ans: D**

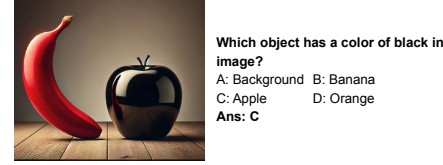
Which object has a color of black in this image?
A: Background   B: Banana
C: Apple    D: Orange
**Ans: C**

Figure 8: **Cases for Object Recognition Task.**

**Color Proportion**

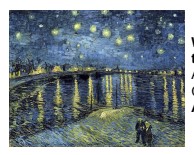
Which is the dominant color in this painting?
A: Blue    B: Yellow
C: Green    D: Orange
**Ans: A**

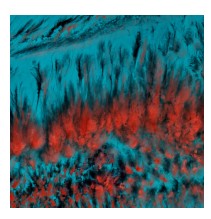
What is closest to the proportion of the color red in the image?
A: 10%    B: 20%
C: 30%    D: 40%
**Ans: C**

Figure 9: **Cases for Color Proportion Task.**

**Color Comparison**

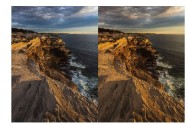
Which photo is warmer in overall color?

A: The left one

B: The right one

**Ans: B**

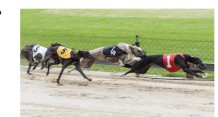
Which dog has the darkest color in the image?
A: No.1    B: No.4
C. No.5    D. No.3
**Ans: A**

Figure 10: **Cases for Color Comparison Task.**

**Color Counting**

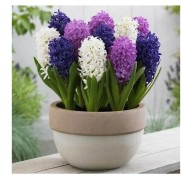
How many different colors of flowers are in this image?
A: 1    B: 2
C: 3    D: 4
**Ans: C**

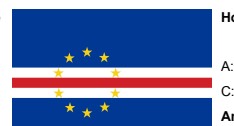
How many colors are there in this flag?
A: 3    B: 4
C: 5    D: 6
**Ans: D**

Figure 11: **Cases for Color Counting Task.**

**Object Counting**

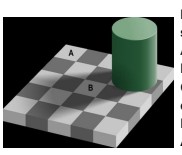
How many striped animals can be seen in this image?
A: 12    B: 11
C: 13    D: 9
E: 10
**Ans: C**

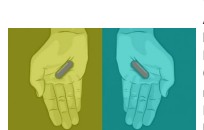
How many green bananas can be seen in this image?
A: 6    B: 7
C. 5    D. 4
E. 0
**Ans: A**

Figure 12: **Cases for Object Counting Task.**

**Color Illusion**

Do the blocks labeled a and b have the same color/shade?
A: No, a is darker
B: Hard to tell without more context
C: Yes, one appears darker due to how our eyes perceive shadows
D: No, b is darker
**Ans: D**

What colors are the two pills?
A: Cannot tell from this image, the colors seem to be shifting?!
B: Both are the exact same shade of gray
C: The left one is bluish-gray and the right one is reddish-grey
D: The left one is reddish-gray and the right one is bluish-grey
**Ans: B**

Figure 13: **Cases for Color Illusion Task.**

**Color Mimicry**

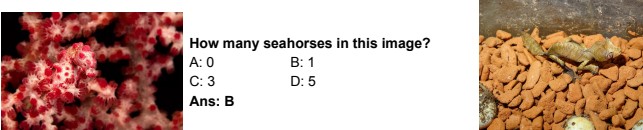

How many seahorses in this image?
A: 0          B: 1
C: 3          D: 5
**Ans: B**

How many leaves in this image?
A: 1          B: 2
C: 3          D: 0
**Ans: D**

Figure 14: **Cases for Color Mimicry Task.**

**Color Blindness**

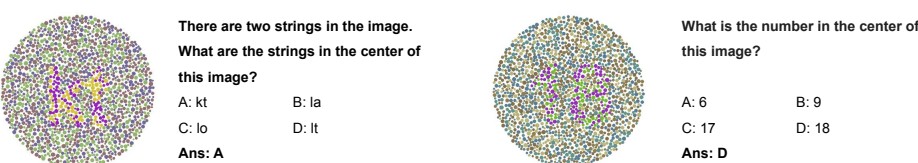

There are two strings in the image.
What are the strings in the center of
this image?
A: kt          B: la
C: lo          D: lt
**Ans: A**

What is the number in the center of
this image?

A: 6          B: 9
C: 17         D: 18
**Ans: D**

Figure 15: **Cases for Color Blindness Task.**

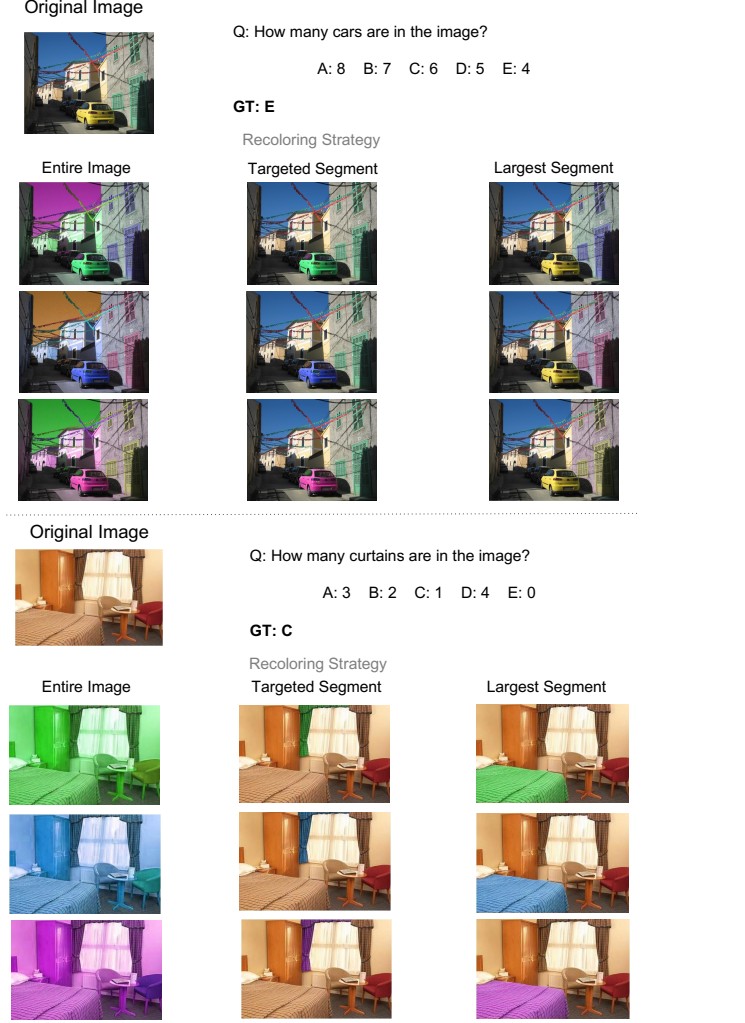

Figure 16: **Cases for Color Robustness Task.**

# E Implementation Details

To further advance our understanding of VLMs' capabilities in color perception, reasoning, and robustness dimensions, we conduct an extensive evaluation of 32 vision-language models (VLMs) spanning a range of large language model (LLM) sizes and architectures. Our evaluation includes state-of-the-art models such as GPT-4o[35], Gemini-2-flash[7], LLaVA-OV[24], LLaVA-NEXT [31], Cambrian[42], InternVL2[5], InternVL2.5[5], Qwen2.5-VL[2], and Eagle[41]. GPT-4o and Gemini-2-flash are used with API calls. We further examine reasoning enhancement via chain-of-thought (CoT) prompting [44], applying it to GPT-4o and Gemini-2-Flash to evaluate how intermediate reasoning steps influence color understanding. Additionally, we include the most recent GPT-o3 on perception and reasoning tasks, which is the most powerful model with a long internal chain-of-thought process. This selection covers a diverse set of architectures, including both proprietary and open-source models, enabling a comprehensive assessment of their reasoning capabilities under different computational constraints.

To ensure a fair comparison, we standardize our experimental setup across models. Open-source models with fewer than 70B parameters are evaluated using a single NVIDIA A100 80GB GPU, while larger models require four NVIDIA A100 80GB GPUs to accommodate their increased memory and computational demands.

# F Evaluation Prompts

> **Instruction Prompt**   You'll be given an image, an instruction and some options. You have to select the correct one. Do not explain your reasoning. Answer with only the letter that corresponds to the correct option. Do not repeat the entire answer.

> **CoT Instruction Prompt**   You'll be given an image, an instruction and some options. You have to select the correct one. Think step by step before answering. Then conclude with the letter that corresponds to the correct option. Make sure the option letter is in the parentheses like (X). Do not include ( or ) in the response except for the answer.

# G Human Evaluation

To assess the degree of alignment between VLMs and human color understanding, we selected a representative subset of COLORBENCH, focusing specifically on color perception and reasoning tasks. The Color Extraction task was excluded from human annotation, as humans are generally not sensitive to fine-grained differences in color codes. Three human participants were recruited, each tasked with completing 50 samples per category. All evaluators responded to the full set of multiple-choice and judgment-oriented questions. We then gathered all responses and conducted statistical analysis on the collected human evaluations.

# H Reasoning Models with Thinking Process

To comprehensively assess the performance of VLMs with the thinking process on COLORBENCH, except for proprietary models with chain-of-thought(CoT) prompt, we additionally conduct experiments with GPT-o3 on perception and reasoning tasks. GPT-o3 is the most recent powerful proprietary VLMs that is trained to think before answering with reinforcement learning. We use the API version of GPT-o3 (2025-04-16) for evaluation. The result is shown in Table 8, together with results of CoT prompting and human evaluation.

The results presented in Table 8 indicate that human evaluators achieve the highest performance across the majority of tasks, except for three specific categories: ***Object Recognition (O'Recog)***, ***Color Proportion (C'Prop)***, and ***Color Comparison (C'Comp)***, where GPT-o3 holds the highest scores. The performance differences between GPT-o3 and human evaluators on ***O'Recog*** and ***C'Comp*** tasks are relatively minor (less than 3%). However, GPT-o3 substantially outperforms both humans and other VLMs on the ***C'Prop*** task, with an advantage exceeding 12%. This significant gap on ***C'Prop*** aligns with expectations, as humans generally exhibit lower sensitivity to precise quantitative measures.

Meanwhile, GPT-o3 benefits from including the capability to utilize analytical tools for precise image assessments and continuous exhaustive visual search [26] to obtain better proportion estimations.

On the remaining tasks, GPT-o3 consistently outperforms GPT-4o (CoT) and Gemini-2-flash (CoT), except for the *Color Blindness (C'Blind)* task, where GPT-o3 trails GPT-4o (CoT) by $3.7\%$. The **C'Blind** task requires VLMs to accurately identify numbers or strings in an image that is composed of colored dots. This task demands capabilities of precise color recognition combined with a holistic spatial perception. One plausible reason for GPT-o3's inferior performance is its longer and more complex reasoning path, which may lead to overthinking. This might cause the model to focus too much on local details or choices of tool, at the expense of the global and intuitive perception needed for this task.

Overall, these findings highlight the relative strengths and weaknesses of current advanced VLMs compared to human evaluators. Importantly, there remains substantial room for improvement in VLM capabilities, as significant performance gaps persist between VLMs and humans, particularly in reasoning-intensive tasks.

Table 8: **Performance of proprietary reasoning models with thinking processes on Color Perception and Reasoning Tasks.** Models are ranked based on their overall performance on color perception and reasoning (P & R Overall) tasks. The best-performing model within the VLM group is highlighted in bold. For human evaluation, any instance that exceeds the performance of all VLMs is also highlighted in bold.

| | Color Perception | | | Color Reasoning | | | | | | | P & R |
|---|---|---|---|---|---|---|---|---|---|---|---|
| | C'Recog | C'Extract | O'Recog | C'Prop | C'Comp | C'Count | O'Count | C'Illu | C'Mimic | C'Blind | Overall |
| *VLMs: Proprietary* | | | | | | | | | | | |
| **GPT-4o (CoT)** | 77.6 | 55.2 | 83.1 | 44.4 | 71.3 | 26.5 | 33.0 | 44.1 | 77.1 | **66.8** | 57.4 |
| **Gemini-2-flash (CoT)** | 82.9 | 56.2 | 88.3 | 58.0 | 68.3 | 43.1 | 38.8 | 40.9 | 75.7 | 60.0 | 59.6 |
| **GPT-o3 (API)** | **84.2** | **57.2** | **92.2** | **71.6** | **82.2** | **46.1** | **45.6** | **58.1** | **80.0** | 63.1 | **66.4** |
| *Human Evaluation* | | | | | | | | | | | |
| **Human Evaluation** | **92.0** | - | 90.1 | 59.6 | 79.8 | **62.0** | **81.3** | **63.0** | **83.8** | **94.0** | - |

# I    Qualitative Analysis of Failure Cases

To gain deeper insights into VLM failures on color-related tasks, we conduct a detailed case analysis using **Qwen2.5-VL-3B** and **7B** models on different tasks. Following the attention visualization methodology of Zhang et al. [49], we focus on instances where the 3B model fails but the 7B model succeeds, allowing a clearer examination of the underlying capability differences. The visualizations of attention maps are shown in Figure 17 to 25.

For Color Perception tasks, we analyze the *Color Recognition* and *Object Recognition* tasks (excluding *Color Extraction*, which contains single-color color images). Our preliminary findings show that only a small number of failures arise from incorrect object localization. In most cases, both models correctly attend to the relevant regions but still produce incorrect predictions. This indicates that VLMs cannot accurately interpret color information, rather than deficiencies in visual grounding for these basic perception tasks.

For Color Reasoning tasks, tasks such as *Color Proportion*, *Color Comparison*, *Color Counting*, and *Color Illusion* require integrating visual information across the entire image without a clear focus point. Attention maps show that both 3B and 7B models exhibit similar focus patterns but generate different answers, implying that the divergence mainly originates from the language reasoning component rather than the visual encoder. For tasks with explicit perception targets, including *Object Counting*, *Color Mimicry*, and *Color Blindness*, both models attend to the correct regions, yet the 3B model often fails to produce accurate predictions. These results reveal that current VLMs remain weak in color interpretability even when their attention is properly aligned.

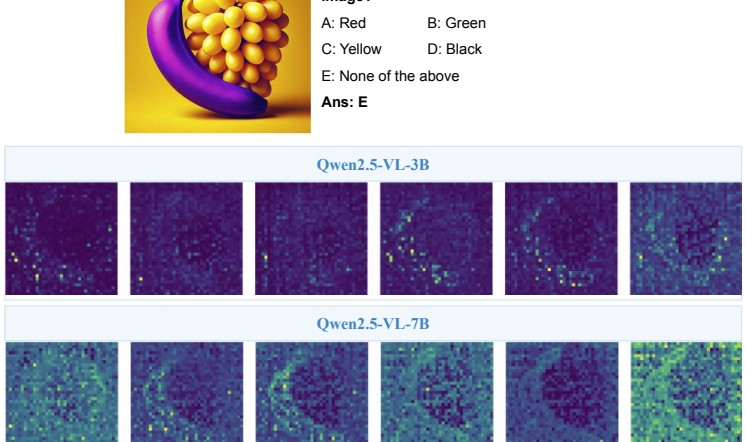

Figure 17: **Visualized Attention Maps for Color Recognition Tasks.**

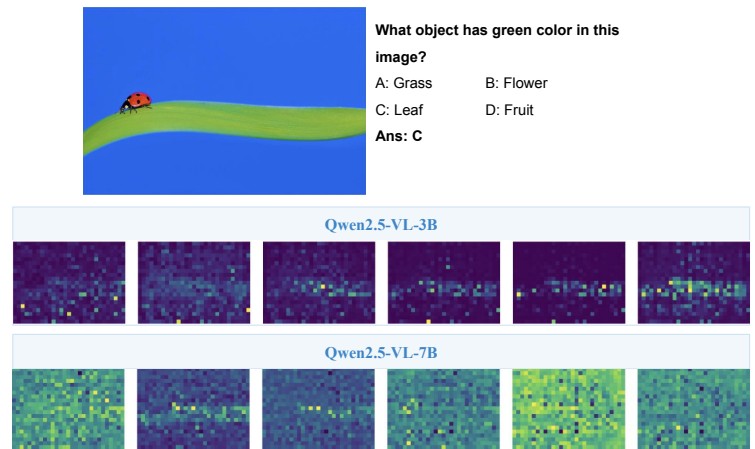

Figure 18: **Visualized Attention Maps for Object Recognition Tasks.**

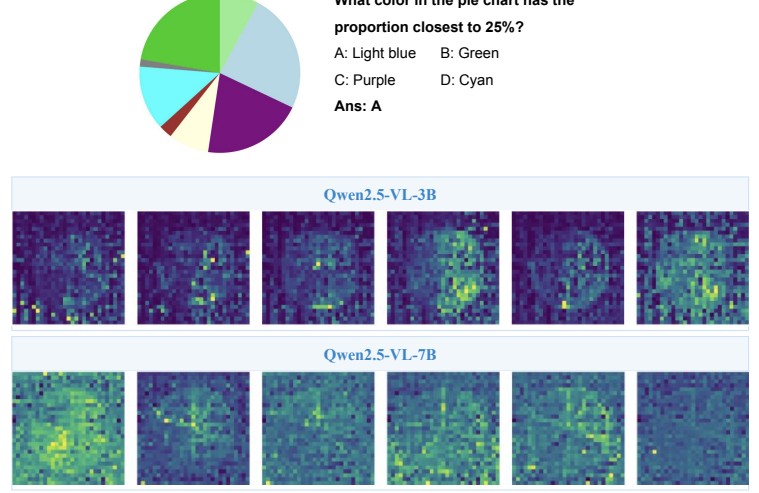

Figure 19: **Visualized Attention Maps for Color Proportion Tasks.**

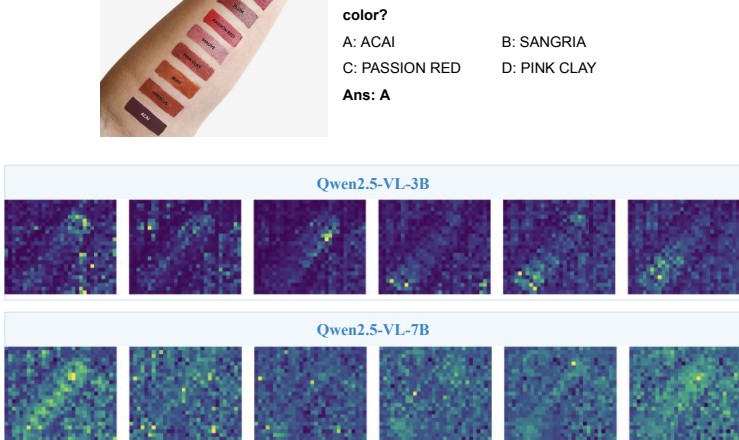

Figure 20: **Visualized Attention Maps for Color Comparison Tasks.**

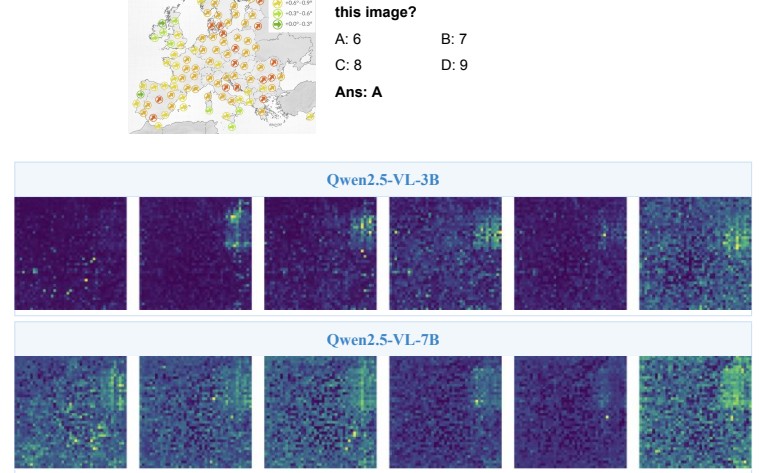

Figure 21: **Visualized Attention Maps for Color Counting Tasks.**

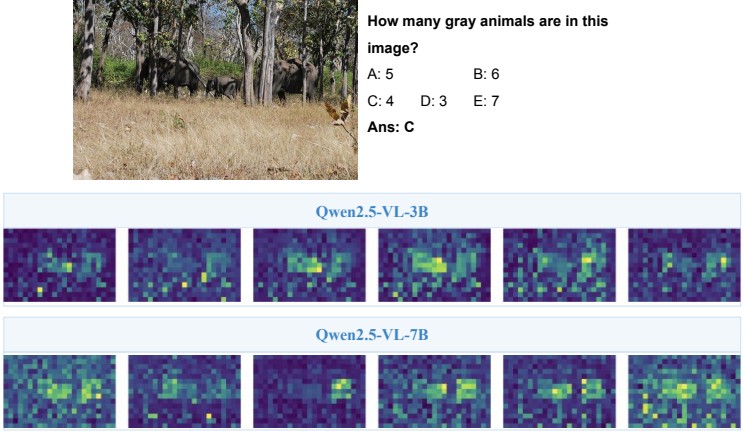

Figure 22: **Visualized Attention Maps for Object Counting Tasks.**

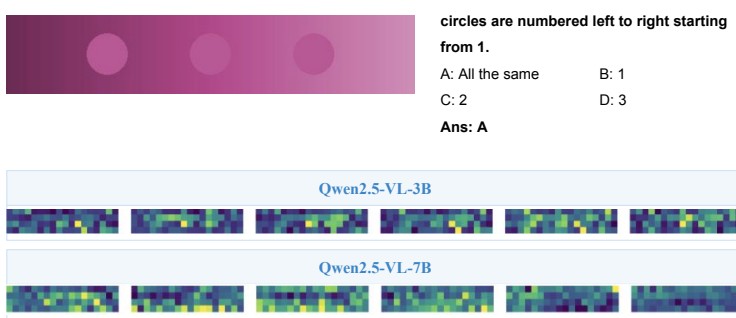

**Which circles has the darkest color? The circles are numbered left to right starting from 1.**

A: All the same        B: 1
C: 2                   D: 3
**Ans: A**

**Qwen2.5-VL-3B**

**Qwen2.5-VL-7B**

Figure 23: **Visualized Attention Maps for Color Illusion Tasks.**

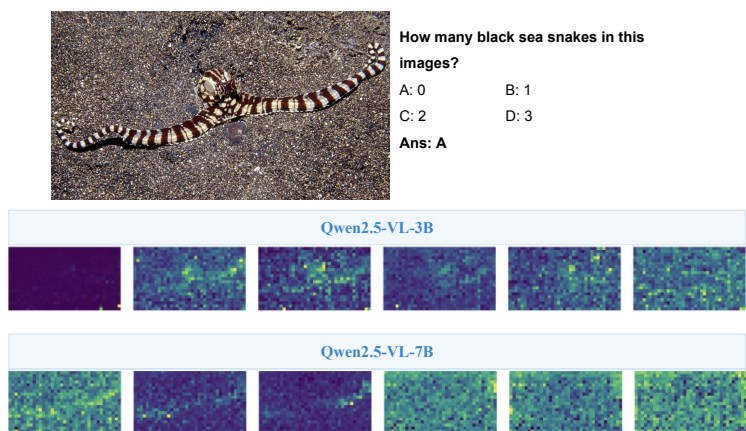

**How many black sea snakes in this images?**

A: 0        B: 1
C: 2        D: 3
**Ans: A**

**Qwen2.5-VL-3B**

**Qwen2.5-VL-7B**

Figure 24: **Visualized Attention Maps for Color Mimicry Tasks.**

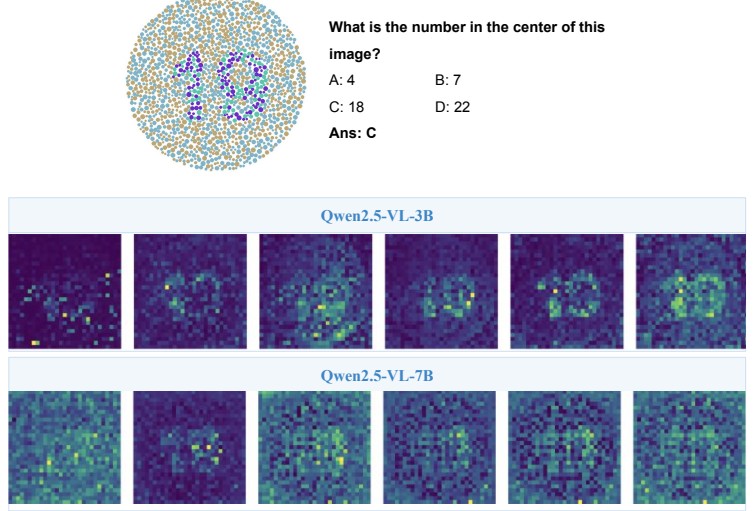

**What is the number in the center of this image?**

A: 4        B: 7
C: 18       D: 22
**Ans: C**

**Qwen2.5-VL-3B**

**Qwen2.5-VL-7B**

Figure 25: **Visualized Attention Maps for Color Blindness Tasks.**

## J    Effect of Different Modalities

To investigate the impact of color information, we compare model performance on RGB versus grayscale images, thereby isolating the role of color within the image modality. To further explore the contribution of the image modality, we also conduct experiments using textual input only (questions and answer choices), where the original input images are substituted with pure black images of identical dimensions.

Table 9: **Average Accuracy (%)** across three input settings (Text-only, Grayscale+Text, RGB+Text) on Color Perception and Reasoning tasks.

| | Color Perception | | | Color Reasoning | | | | | | | P & R |
|---|---|---|---|---|---|---|---|---|---|---|---|
| | C'Recog | C'Extract | O'Recog | C'Prop | C'Comp | C'Count | O'Count | C'Illu | C'Mimic | C'Blind | Overall |
| *VLMs: < 7B* | | | | | | | | | | | |
| **Text-only** | 29.2 | 30.6 | 31.6 | 29.6 | 35.3 | **24.5** | 20.6 | 35.5 | 41.7 | 23.4 | 29.3 |
| **Gray+Text** | 25.9 | 33.5 | 42.7 | 29.1 | 37.1 | 23.2 | 23.3 | **42.4** | **53.7** | 23.0 | 32.1 |
| **RGB+Text** | **55.3** | **35.7** | **63.6** | **37.3** | **42.4** | 22.5 | **26.1** | 37.5 | 50.6 | **25.0** | **37.4** |
| *VLMs: 7B − 8B* | | | | | | | | | | | |
| **Text-only** | 23.7 | 35.4 | 32.3 | 20.6 | 29.7 | 18.4 | 19.3 | 36.7 | 36.9 | 21.1 | 26.7 |
| **Gray+Text** | 25.2 | 35.7 | 46.0 | 27.8 | 41.3 | 22.2 | 27.5 | **48.2** | **58.7** | **23.6** | 34.2 |
| **RGB+Text** | **60.4** | **42.4** | **73.0** | **41.8** | **49.1** | **22.7** | **32.7** | 41.5 | 50.0 | 23.4 | **41.1** |
| *VLMs: 10B − 30B* | | | | | | | | | | | |
| **Text-only** | 26.9 | 33.6 | 32.8 | 25.0 | 34.7 | **26.5** | 22.3 | 38.2 | 40.0 | 18.9 | 28.9 |
| **Gray+Text** | 26.8 | 37.9 | 46.8 | 22.5 | 46.5 | 22.4 | 30.1 | **43.0** | **60.3** | 26.0 | 35.0 |
| **RGB+Text** | **68.4** | **41.5** | **79.7** | **43.0** | **51.3** | 25.3 | **34.4** | 33.8 | 55.4 | **26.6** | **43.2** |
| *VLMs: 30B − 70B* | | | | | | | | | | | |
| **Text-only** | 28.9 | 36.5 | 31.8 | 16.3 | 29.0 | 15.4 | 16.3 | 42.7 | 33.6 | 15.9 | 25.6 |
| **Gray+Text** | 28.7 | 42.1 | 51.2 | 26.3 | 49.9 | 24.3 | 25.6 | **48.8** | **65.1** | 22.7 | 36.7 |
| **RGB+Text** | **73.4** | **48.8** | **81.6** | **49.5** | **55.2** | **24.7** | **37.3** | 36.1 | 61.1 | **25.5** | **46.2** |
| *VLMs: > 70B* | | | | | | | | | | | |
| **Text-only** | 26.0 | 47.4 | 35.7 | 20.9 | 36.9 | 21.6 | 24.0 | 35.8 | 33.9 | 21.8 | 29.8 |
| **Gray+Text** | 25.3 | 40.9 | 54.6 | 25.3 | 51.0 | 21.8 | 28.6 | **44.6** | **54.3** | 26.1 | 36.1 |
| **RGB+Text** | **73.4** | **54.7** | **82.5** | **45.6** | **62.4** | **26.7** | **39.6** | 33.9 | 53.9 | **29.6** | **47.6** |

Table 9 presents the average accuracy across models grouped by LLM size. The result demonstrates that removing the visual modality (*text-only setting*) leads to the lowest performance across the majority of tasks. The performance differences among the three input settings allow us to disentangle the impact of textual input, image context (excluding color), and color information itself.

Notably, in tasks such as ***Color Recognition*** and ***Object Recognition***, the performance gap between text-only and grayscale experiments is relatively small, whereas both are significantly outperformed by the RGB input setting. This suggests that color cues play a substantially more important role than either contextual visual or textual information in these tasks.

## K    Fine-tuning Experiments on ColorBench

We conduct a series of fine-tuning experiments to investigate model adaptation on specialized color-centric tasks. These experiments leverage three synthetic datasets designed for ***Color Extraction***, ***Color Illusion***, and ***Color Blindness***. Using our synthetic data generation pipeline, we curate dedicated training sets for this purpose, with sample counts summarized in Table 10.

Table 10: **Number of synthetic samples generated** for fine-tuning experiments.

| Task | Number of Samples |
|---|---|
| Color Extraction | 2400 |
| Color Illusion | 2400 |
| Color Blindness | 2280 |

To systematically assess the influence of different model components, we perform a comprehensive ablation study on **Qwen2.5-VL-3B** and **Qwen2.5-VL-7B** with the following settings:

- MLP only

- Vision encoder only

- MLP + Vision encoder (jointly)

- LLM (LoRA) only

- LLM (LoRA) + MLP

- LLM (LoRA) + Vision encoder

- LLM (LoRA) + MLP + Vision encoder (jointly)

For configurations involving the LLM, we adopt the LoRA approach to update a subset of its parameters, while the remaining modules are fully fine-tuned.

Table 11: **Accuracy (%)** of Qwen2.5-VL (3B and 7B) under different training strategies across ColorBench tasks. Bold numbers indicate the best results within each model group.

| Model | Trainable Modules | | | Color Perception | | | Color Reasoning | | | | | | | P&R |
|---|---|---|---|---|---|---|---|---|---|---|---|---|---|---|
| | LLM (LoRA) | MLP | Vision | C'Recog | C'Extract | O'Recog | C'Prop | C'Comp | C'Count | O'Count | C'Illu | C'Mimic | C'Blind | Overall |
| Qwen2.5-3B | | | | 72.4 | 38.5 | 74.0 | 43.8 | 48.5 | 22.6 | 25.2 | 43.0 | 45.7 | 24.2 | 41.1 |
| | | ✓ | | 71.1 | 53.1 | 75.3 | **50.0** | 49.5 | 22.5 | 26.2 | 45.2 | 44.3 | 25.5 | 43.6 |
| | | | ✓ | 73.7 | 53.1 | **79.2** | 46.3 | 45.5 | **29.4** | 27.2 | 48.4 | 47.1 | 25.5 | 44.4 |
| | | ✓ | ✓ | **75.0** | 56.3 | 75.3 | 47.5 | 49.5 | 28.4 | 25.2 | 46.2 | 47.1 | 28.0 | 45.2 |
| | ✓ | | | 71.1 | 75.0 | 70.1 | 45.0 | 51.5 | 26.5 | 27.2 | 45.2 | 47.1 | 27.4 | 46.2 |
| | ✓ | ✓ | | 69.7 | **77.1** | 74.0 | 40.0 | **53.5** | 23.5 | **32.0** | 51.6 | 45.7 | **37.6** | **48.8** |
| | ✓ | | ✓ | 71.1 | 75.0 | 71.4 | 46.3 | 49.5 | 25.5 | 27.2 | 49.4 | 48.6 | 31.4 | 46.7 |
| | ✓ | ✓ | ✓ | 72.4 | 75.0 | 71.4 | 45.0 | 51.5 | 24.3 | **32.0** | 46.2 | **50.0** | 28.0 | 47.1 |
| Qwen2.5-7B | | | | 76.3 | 49.0 | **84.4** | 47.5 | 52.5 | 19.6 | 34.0 | 44.1 | **55.7** | 28.7 | 46.2 |
| | | ✓ | | 72.4 | 42.7 | **84.4** | 42.5 | 59.4 | 20.6 | 29.1 | 47.1 | 50.0 | 28.7 | 45.2 |
| | | | ✓ | 77.6 | 59.4 | 81.8 | 47.5 | 56.4 | **25.5** | 29.1 | 51.6 | 50.0 | **35.6** | 51.2 |
| | | ✓ | ✓ | **78.9** | 61.5 | 80.5 | 41.3 | 55.4 | 20.6 | 29.1 | 47.3 | 48.6 | 30.1 | 47.7 |
| | ✓ | | | 75.0 | 78.1 | 83.1 | **51.3** | **60.4** | 21.6 | **35.0** | 52.7 | 54.3 | **35.6** | **52.4** |
| | ✓ | ✓ | | 72.4 | 82.3 | 83.1 | **51.3** | 57.4 | 19.6 | 30.1 | 51.6 | 52.9 | 33.1 | 51.2 |
| | ✓ | | ✓ | 75.0 | **83.3** | 83.1 | 45.0 | 56.4 | 15.7 | 30.1 | **53.8** | 54.3 | 33.1 | 51.5 |
| | ✓ | ✓ | ✓ | 77.6 | 82.3 | 83.1 | 50.0 | 55.5 | 23.3 | 31.1 | 52.7 | **55.7** | 33.1 | 51.7 |

The evaluation results with finetuned VLMs are shown in Table 11. Overall, models that include LoRA fine-tuning on the LLM component consistently outperform those without it, exhibiting a substantial improvement in overall accuracy. Importantly, the improvements are not confined to the directly targeted tasks (*Color Extraction*, *Color Illusion*, *Color Blindness*). These experiments show that fine-tuning the model on part of tasks also produces notable gains on some ancillary reasoning tasks, including *Color Proportion*, and *Color Comparison*.

However, the transfer of knowledge is not universally positive. Certain tasks demonstrated limited or even negative performance transfer, indicating that fine-tuning exclusively on specialized color objectives does not guarantee generalization across the full spectrum of color perception and reasoning. This finding underscores that while targeted training enhances specialized abilities, a balanced and robust performance profile necessitates the inclusion of more diverse data and training objectives.

# L   More Visualizations

## L.1   VLM Size & Model Performance for Each Task

Figure 26 to 35 present detailed correlations between the log-scaled sizes of **VLM parameters** and the performance metrics for each task of Perception and Reasoning Categories. Deeper color represents higher accuracy. Each line represents a model family with the sizes growing from small to large. This visualization clearly shows the correlation between performances and model sizes, larger model leads to higher performance.

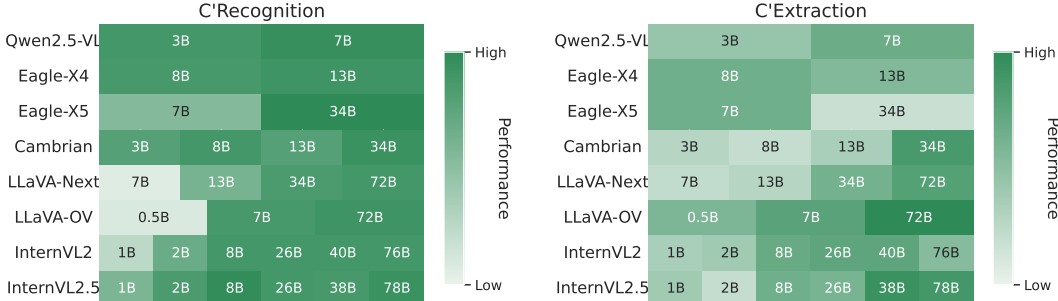

Figure 26: **Heatmap for Color Recognition.**  Figure 27: **Heatmap for Color Extraction.**

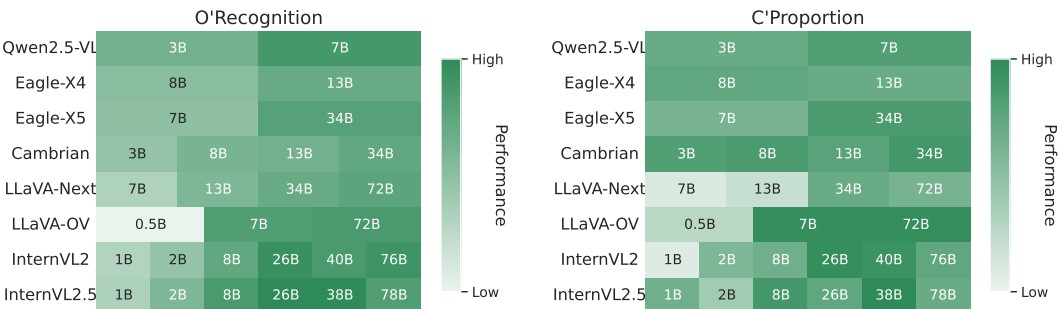

Figure 28: **Heatmap for Object Recognition.**  Figure 29: **Heatmap for Color Proportion.**

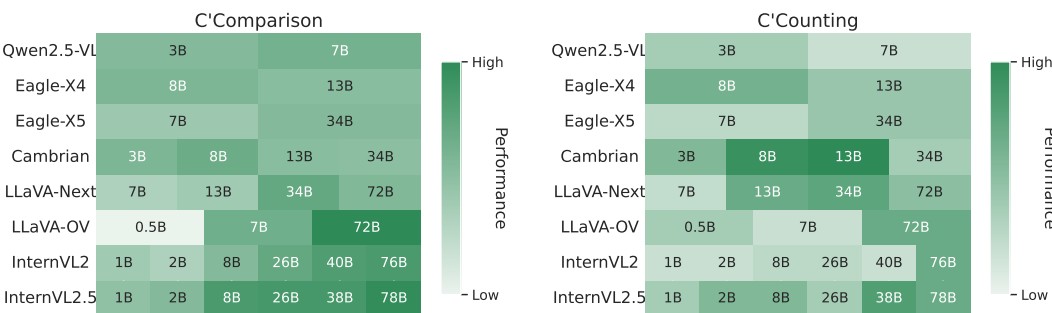

Figure 30: **Heatmap for Color Comparison.**  Figure 31: **Heatmap for Color Counting.**

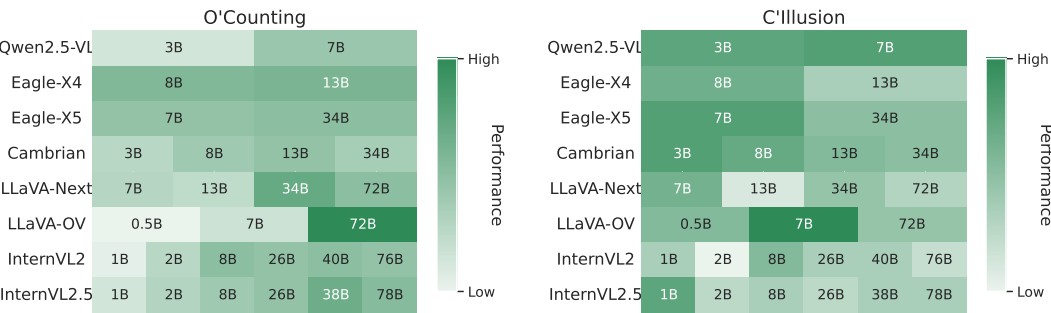

Figure 32: **Heatmap for Object Counting.**  Figure 33: **Heatmap for Color Illusion.**

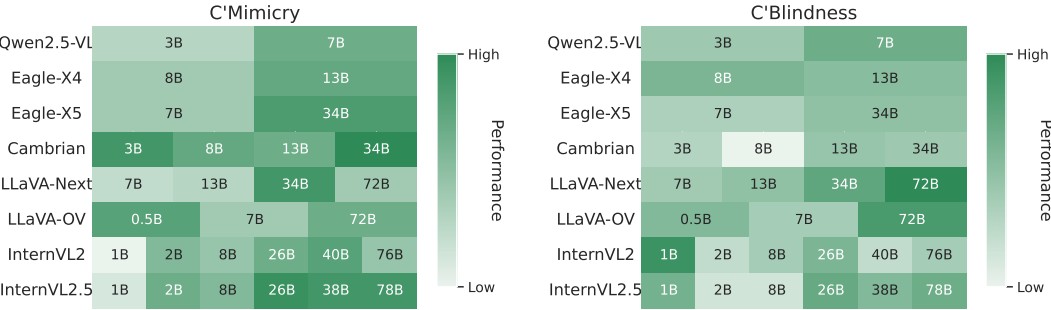

Figure 34: **Heatmap for Color Mimicry.**    Figure 35: **Heatmap for Color Blindness.**

## L.2 Vision Size & Model Performance for Each Task

Figure 36 to 40 show detailed correlations between the log-scaled sizes of **vision encoders** and the performance metrics for each task of Perception and Reasoning Categories. Colors represent different model families. Models that have the same vision encoder sizes but with different LLM sizes are plotted as different points. Given that the majority of Vision-Language Models (VLMs) utilize a singular type of vision encoder, and that the sizes of these encoders generally range between 300-400M, it becomes challenging to assess the scaling effects within vision encoders.

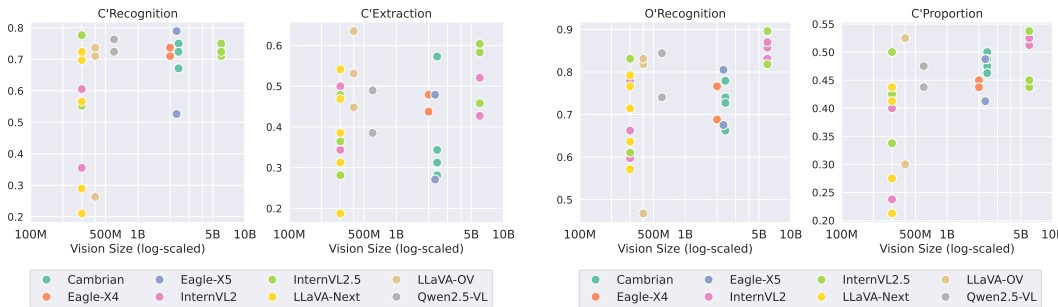

Figure 36: **The scatter plot for Color Recognition and Color Extraction.**

Figure 37: **The scatter plot for Object Recognition and Color Proportion.**

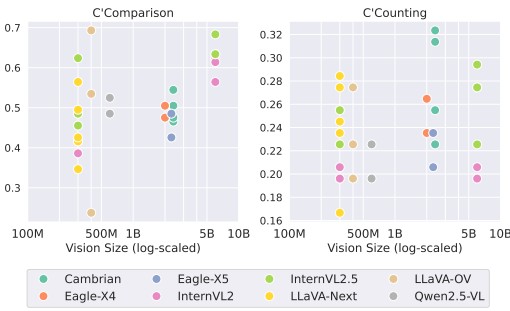
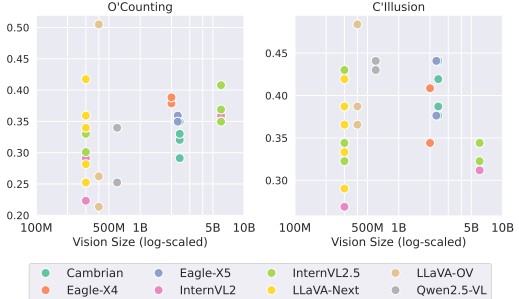

Figure 38: **The scatter plot for Color Comparison and Color Counting.**

Figure 39: **The scatter plot for Object Counting and Color Illusion.**

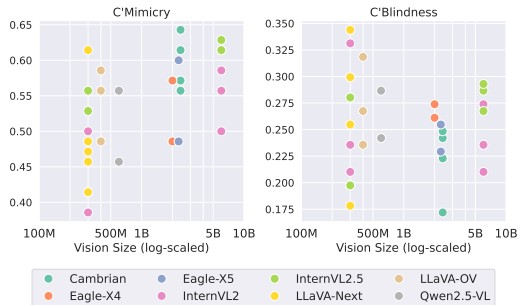

Figure 40: **The scatter plot for Color Mimicry and Color Blindness.**

## L.3    Performance for Each Model Family on Each Task

Figures 41 to 47 illustrate task performance across different models within the same model families. In general, models with more parameters tend to perform better on the majority of tasks.

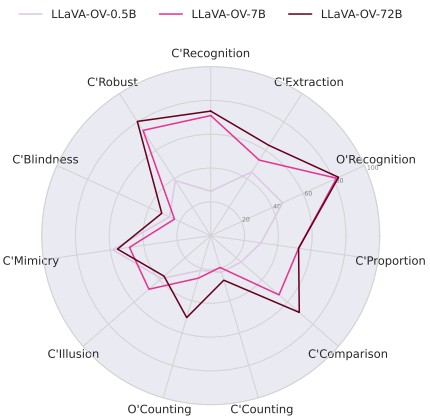

Figure 41: **Performance of LLaVA-OV models.**

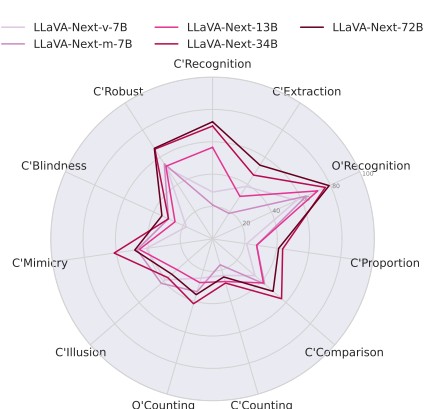

Figure 42: **Performance of LLaVA-NEXT models.**

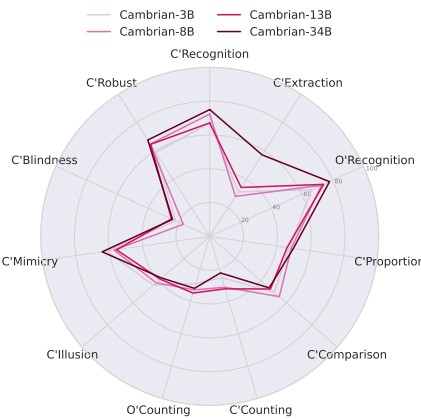

Figure 43: **Performance of Cambrian models.**

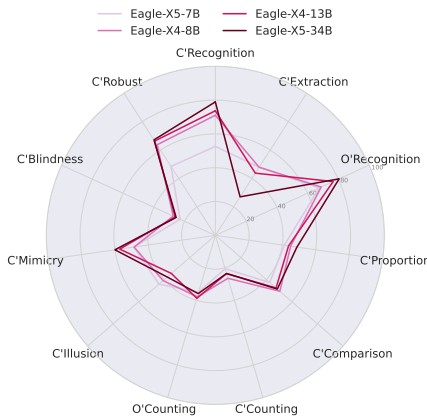

Figure 44: **Performance of Eagle models.**

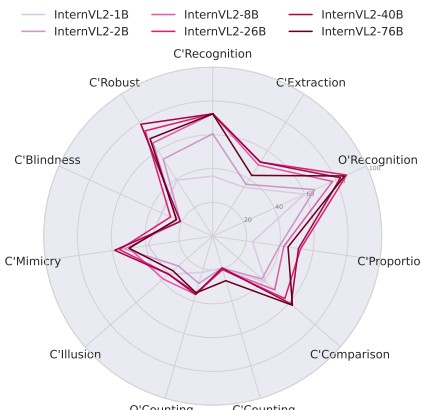

Figure 45: **Performance of InternVL2 models.**

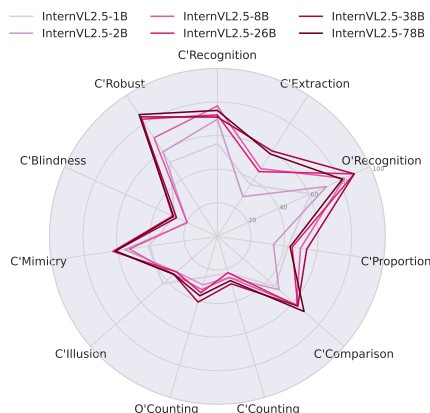

Figure 46: **Performance of In­ternVL2.5 models.**

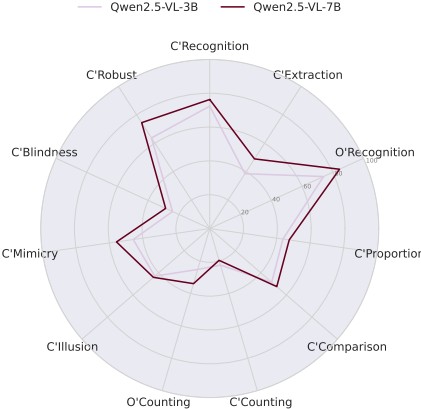

Figure 47: **Performance of Qwen2.5 models.**

# M  Samples Cases

## M.1  Effect of CoT

In this section, we present cases that the answers are influenced by adding reasoning steps for each task. For most of the tasks in COLORBENCH, adding reasoning steps can significantly improve the model performances. The samples cases of Perception and Reasoning categories are shown in Figure 48 to Figure 57. Case for Robustness category is shown in Figure 58.

---

### Color Recognition

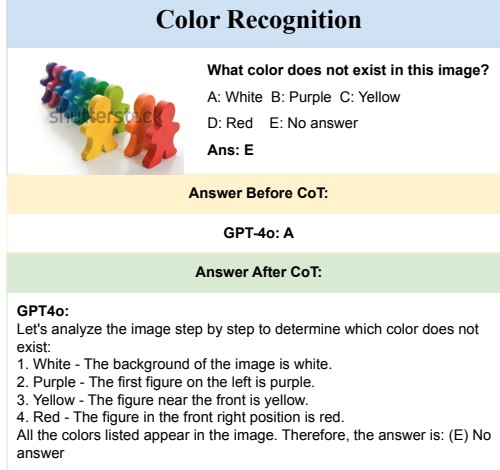

**What color does not exist in this image?**
A: White  B: Purple  C: Yellow
D: Red    E: No answer
**Ans: E**

**Answer Before CoT:**

**GPT-4o: A**

**Answer After CoT:**

GPT4o:
Let's analyze the image step by step to determine which color does not exist:
1. White - The background of the image is white.
2. Purple - The first figure on the left is purple.
3. Yellow - The figure near the front is yellow.
4. Red - The figure in the front right position is red.
All the colors listed appear in the image. Therefore, the answer is: (E) No answer

Figure 48: **Case with CoT for Color Recognition task**.

---

### Color Extraction

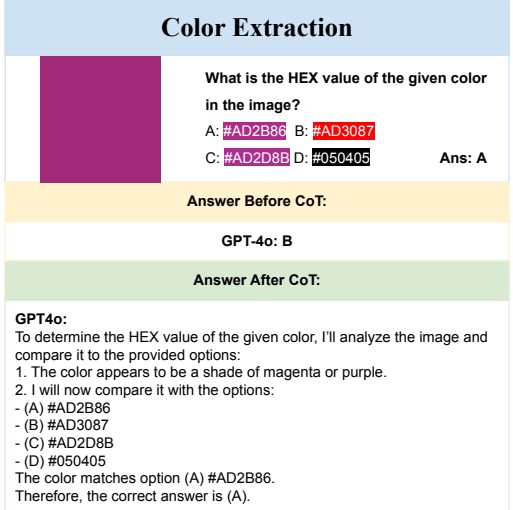

**What is the HEX value of the given color in the image?**
A: #AD2B86  B: #AD3087
C: #AD2D8B  D: #050405        **Ans: A**

**Answer Before CoT:**

**GPT-4o: B**

**Answer After CoT:**

GPT4o:
To determine the HEX value of the given color, I'll analyze the image and compare it to the provided options:
1. The color appears to be a shade of magenta or purple.
2. I will now compare it with the options:
- (A) #AD2B86
- (B) #AD3087
- (C) #AD2D8B
- (D) #050405
The color matches option (A) #AD2B86.
Therefore, the correct answer is (A).

Figure 49: **Case with CoT for Color Extraction task**. Option backgrounds correspond to their color codes.

---

### Object Recognition

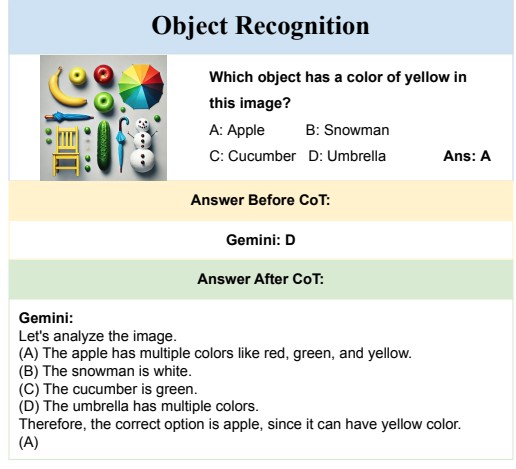

**Which object has a color of yellow in this image?**
A: Apple        B: Snowman
C: Cucumber  D: Umbrella     **Ans: A**

**Answer Before CoT:**

**Gemini: D**

**Answer After CoT:**

Gemini:
Let's analyze the image.
(A) The apple has multiple colors like red, green, and yellow.
(B) The snowman is white.
(C) The cucumber is green.
(D) The umbrella has multiple colors.
Therefore, the correct option is apple, since it can have yellow color.
(A)

Figure 50: **Case with CoT for Object Recognition task**.

---

### Color Proportion

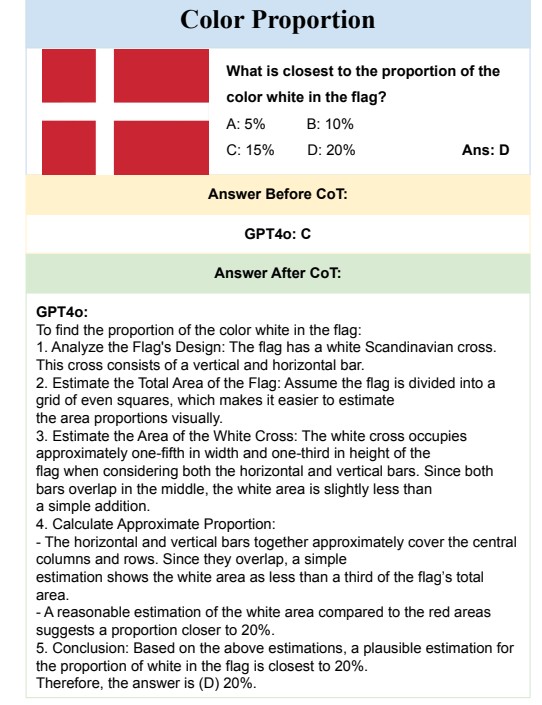

**What is closest to the proportion of the color white in the flag?**
A: 5%        B: 10%
C: 15%       D: 20%          **Ans: D**

**Answer Before CoT:**

**GPT4o: C**

**Answer After CoT:**

GPT4o:
To find the proportion of the color white in the flag:
1. Analyze the Flag's Design: The flag has a white Scandinavian cross. This cross consists of a vertical and horizontal bar.
2. Estimate the Total Area of the Flag: Assume the flag is divided into a grid of even squares, which makes it easier to estimate the area proportions visually.
3. Estimate the Area of the White Cross: The white cross occupies approximately one-fifth in width and one-third in height of the flag when considering both the horizontal and vertical bars. Since both bars overlap in the middle, the white area is slightly less than a simple addition.
4. Calculate Approximate Proportion:
- The horizontal and vertical bars together approximately cover the central columns and rows. Since they overlap, a simple estimation shows the white area as less than a third of the flag's total area.
- A reasonable estimation of the white area compared to the red areas suggests a proportion closer to 20%.
5. Conclusion: Based on the above estimations, a plausible estimation for the proportion of white in the flag is closest to 20%.
Therefore, the answer is (D) 20%.

Figure 51: **Case with CoT for Color Proportion task**.

## Color Comparison

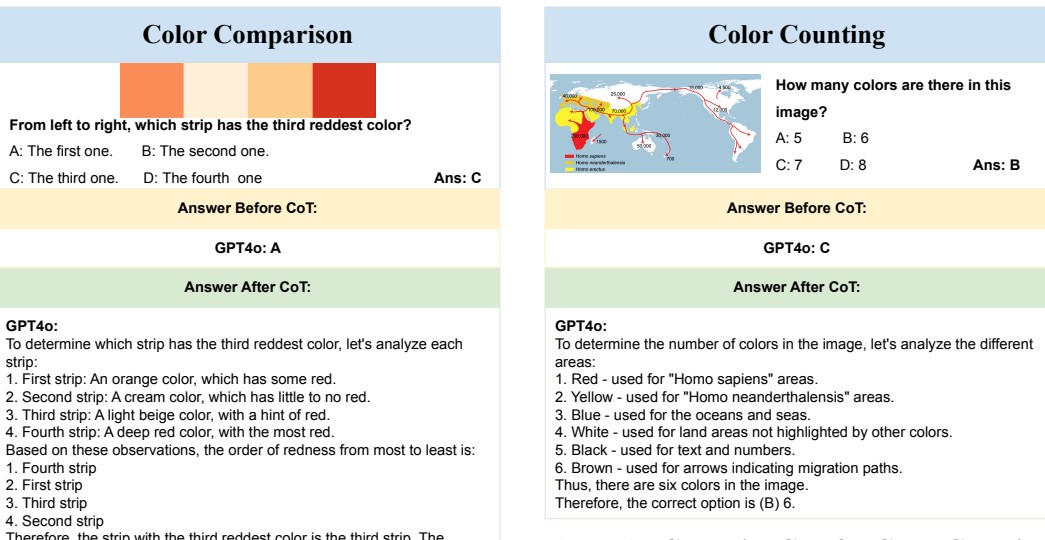

**From left to right, which strip has the third reddest color?**

A: The first one.    B: The second one.

C: The third one.    D: The fourth one          **Ans: C**

**Answer Before CoT:**

**GPT4o: A**

**Answer After CoT:**

**GPT4o:**
To determine which strip has the third reddest color, let's analyze each strip:
1. First strip: An orange color, which has some red.
2. Second strip: A cream color, which has little to no red.
3. Third strip: A light beige color, with a hint of red.
4. Fourth strip: A deep red color, with the most red.
Based on these observations, the order of redness from most to least is:
1. Fourth strip
2. First strip
3. Third strip
4. Second strip
Therefore, the strip with the third reddest color is the third strip. The correct option is (C)

Figure 52: **Case with CoT for Color Comparison task**.

## Color Counting

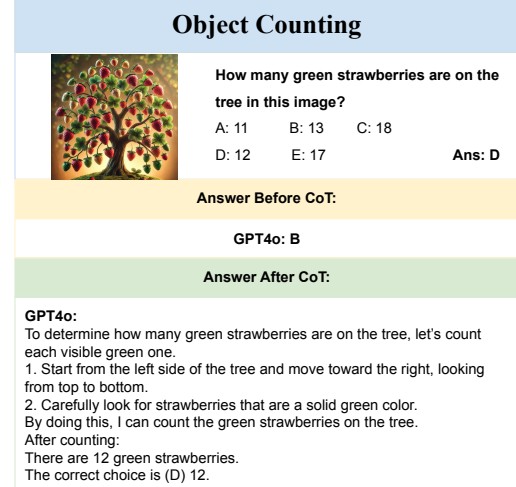

**How many colors are there in this image?**

A: 5        B: 6

C: 7        D: 8          **Ans: B**

**Answer Before CoT:**

**GPT4o: C**

**Answer After CoT:**

**GPT4o:**
To determine the number of colors in the image, let's analyze the different areas:
1. Red - used for "Homo sapiens" areas.
2. Yellow - used for "Homo neanderthalensis" areas.
3. Blue - used for the oceans and seas.
4. White - used for land areas not highlighted by other colors.
5. Black - used for text and numbers.
6. Brown - used for arrows indicating migration paths.
Thus, there are six colors in the image.
Therefore, the correct option is (B) 6.

Figure 53: **Case with CoT for Color Counting task**.

## Object Counting

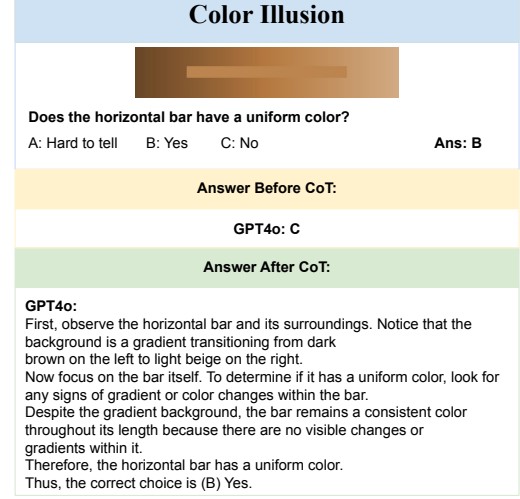

**How many green strawberries are on the tree in this image?**

A: 11        B: 13        C: 18

D: 12        E: 17          **Ans: D**

**Answer Before CoT:**

**GPT4o: B**

**Answer After CoT:**

**GPT4o:**
To determine how many green strawberries are on the tree, let's count each visible green one.
1. Start from the left side of the tree and move toward the right, looking from top to bottom.
2. Carefully look for strawberries that are a solid green color.
By doing this, I can count the green strawberries on the tree.
After counting:
There are 12 green strawberries.
The correct choice is (D) 12.

Figure 54: **Case with CoT for Object Counting task**.

## Color Illusion

**Does the horizontal bar have a uniform color?**

A: Hard to tell      B: Yes      C: No          **Ans: B**

**Answer Before CoT:**

**GPT4o: C**

**Answer After CoT:**

**GPT4o:**
First, observe the horizontal bar and its surroundings. Notice that the background is a gradient transitioning from dark brown on the left to light beige on the right.
Now focus on the bar itself. To determine if it has a uniform color, look for any signs of gradient or color changes within the bar.
Despite the gradient background, the bar remains a consistent color throughout its length because there are no visible changes or gradients within it.
Therefore, the horizontal bar has a uniform color.
Thus, the correct choice is (B) Yes.

Figure 55: **Case with CoT for Color Illusion task**.

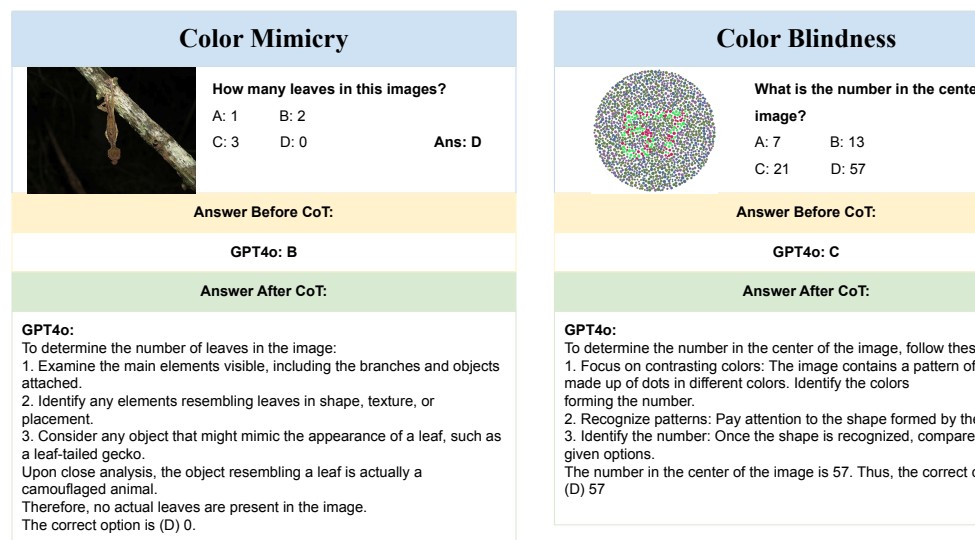

## Color Mimicry

**How many leaves in this images?**

A: 1    B: 2

C: 3    D: 0    **Ans: D**

**Answer Before CoT:**

**GPT4o: B**

**Answer After CoT:**

**GPT4o:**
To determine the number of leaves in the image:
1. Examine the main elements visible, including the branches and objects attached.
2. Identify any elements resembling leaves in shape, texture, or placement.
3. Consider any object that might mimic the appearance of a leaf, such as a leaf-tailed gecko.
Upon close analysis, the object resembling a leaf is actually a camouflaged animal.
Therefore, no actual leaves are present in the image.
The correct option is (D) 0.

Figure 56: **Case with CoT for Color Mimicry task**.

## Color Blindness

**What is the number in the center of this image?**

A: 7    B: 13

C: 21    D: 57    **Ans: D**

**Answer Before CoT:**

**GPT4o: C**

**Answer After CoT:**

**GPT4o:**
To determine the number in the center of the image, follow these steps:
1. Focus on contrasting colors: The image contains a pattern of numbers made up of dots in different colors. Identify the colors forming the number.
2. Recognize patterns: Pay attention to the shape formed by these colors.
3. Identify the number: Once the shape is recognized, compare it with the given options.
The number in the center of the image is 57. Thus, the correct choice is: (D) 57

Figure 57: **Case with CoT for Color Blindness task**.

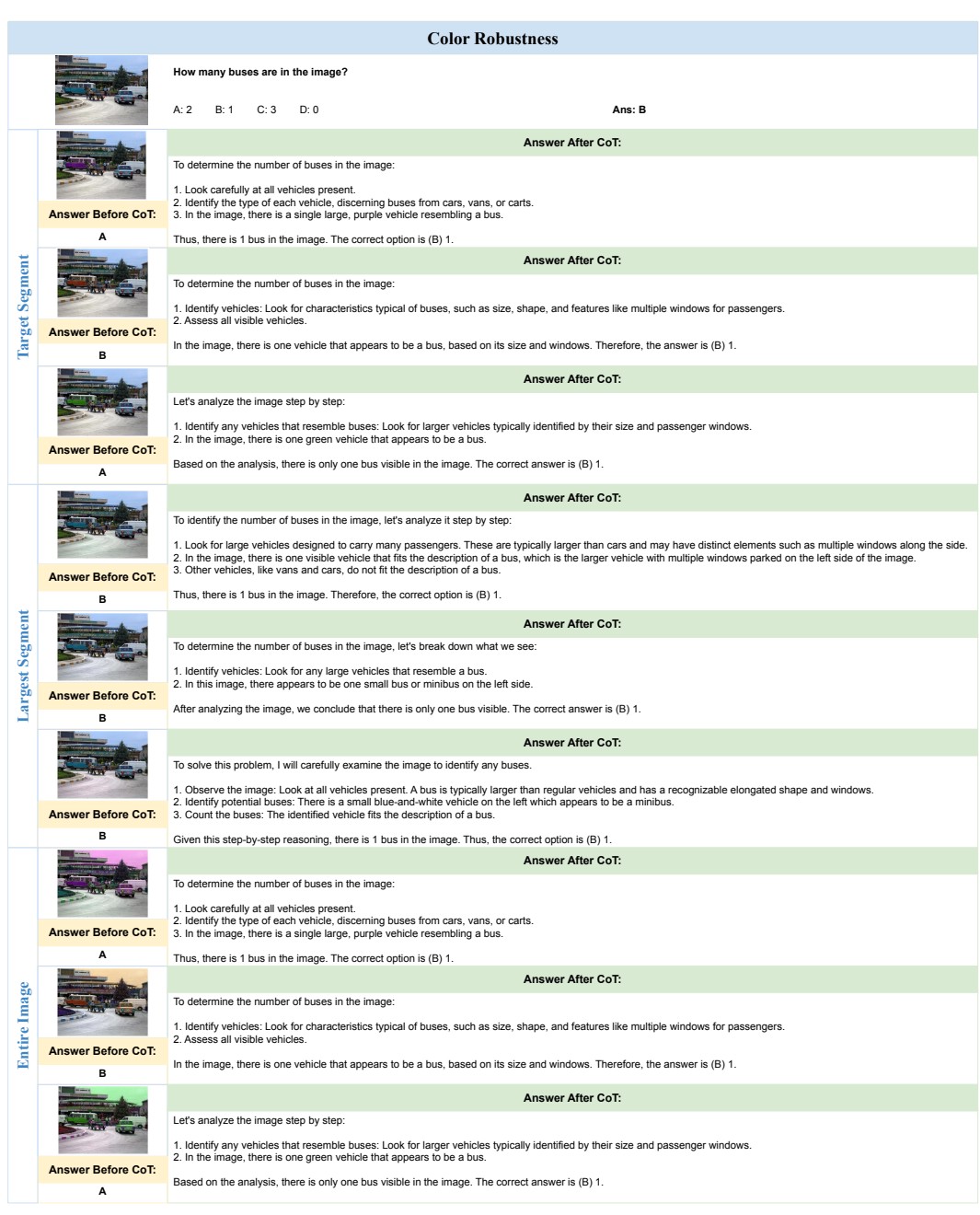

Figure 58: **Case with CoT for Color Robustness task.**

However, for Color Recognition and Object Recognition tasks, the improvement of involving slow thinking is limited, as these two tasks heavily rely on the accurate cognition of the vision encoder. The sample cases are shown in Figure 59 and 60. For Color Illusion task, adding reasoning steps causes the model to focus more on the misleading environment and the relationship between the environment and the foreground objects. This thinking negatively influences the model performance. A sample case is shown by Figure 61.

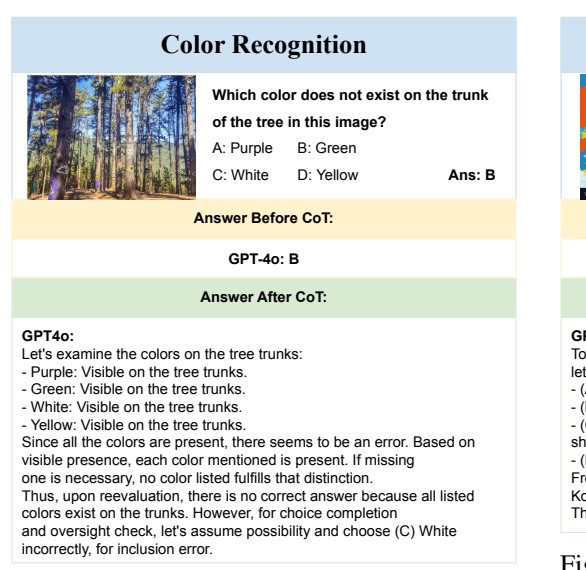

**Color Recognition**

Which color does not exist on the trunk of the tree in this image?
A: Purple     B: Green
C: White      D: Yellow                    **Ans: B**

**Answer Before CoT:**

GPT-4o: B

**Answer After CoT:**

GPT4o:
Let's examine the colors on the tree trunks:
- Purple: Visible on the tree trunks.
- Green: Visible on the tree trunks.
- White: Visible on the tree trunks.
- Yellow: Visible on the tree trunks.
Since all the colors are present, there seems to be an error. Based on visible presence, each color mentioned is present. If missing one is necessary, no color listed fulfills that distinction.
Thus, upon reevaluation, there is no correct answer because all listed colors exist on the trunks. However, for choice completion and oversight check, let's assume possibility and choose (C) White incorrectly, for inclusion error.

Figure 59: **Case that CoT results in an incorrect answer for Color Recognition task**.

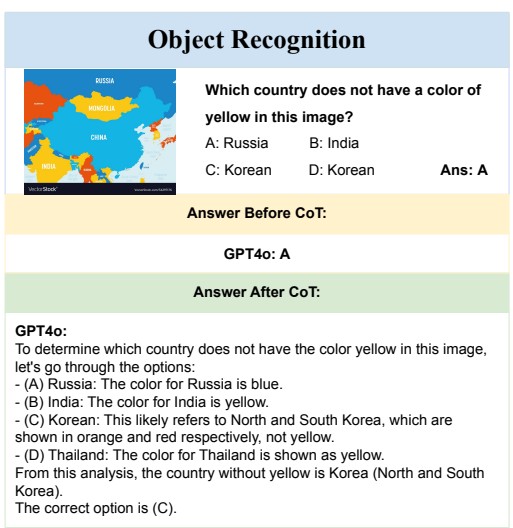

**Object Recognition**

Which country does not have a color of yellow in this image?
A: Russia       B: India
C: Korean       D: Korean                  **Ans: A**

**Answer Before CoT:**

GPT4o: A

**Answer After CoT:**

GPT4o:
To determine which country does not have the color yellow in this image, let's go through the options:
- (A) Russia: The color for Russia is blue.
- (B) India: The color for India is yellow.
- (C) Korean: This likely refers to North and South Korea, which are shown in orange and red respectively, not yellow.
- (D) Thailand: The color for Thailand is shown as yellow.
From this analysis, the country without yellow is Korea (North and South Korea).
The correct option is (C).

Figure 60: **Case that CoT results in an incorrect answer for Object Recognition task**.

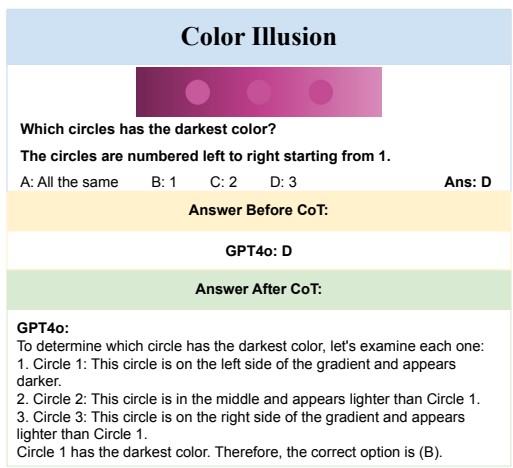

**Color Illusion**

Which circles has the darkest color?

The circles are numbered left to right starting from 1.

A: All the same      B: 1      C: 2      D: 3          **Ans: D**

**Answer Before CoT:**

GPT4o: D

**Answer After CoT:**

GPT4o:
To determine which circle has the darkest color, let's examine each one:
1. Circle 1: This circle is on the left side of the gradient and appears darker.
2. Circle 2: This circle is in the middle and appears lighter than Circle 1.
3. Circle 3: This circle is on the right side of the gradient and appears lighter than Circle 1.
Circle 1 has the darkest color. Therefore, the correct option is (B).

Figure 61: **Case that CoT results in an incorrect answer for Color Illusion task**.

## M.2 Effect of Grayscale

For most of the tasks in COLORBENCH, colors are critical clues for VLMs to generate the answers. We highlight these cases in Figure 62 to 69.

However, for Color Illusion and Color Mimicry tasks, color clues might mislead VLMs to wrong answers, as shown in Figure 70 and 71.

**Color Recognition**

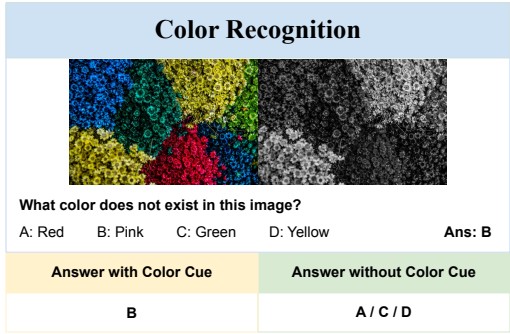

What color does not exist in this image?

A: Red    B: Pink    C: Green    D: Yellow    **Ans: B**

| Answer with Color Cue | Answer without Color Cue |
| --- | --- |
| B | A / C / D |

Figure 62: **Color clues play as a critical role for Color Recognition task**.

**Color Extraction**

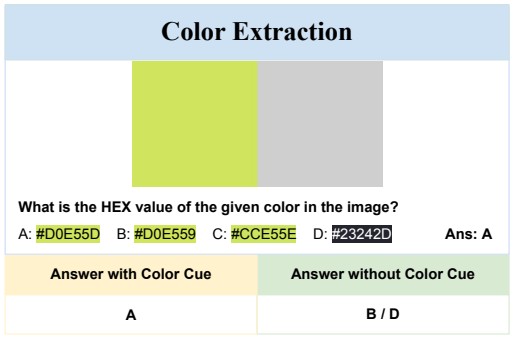

What is the HEX value of the given color in the image?

A: #D0E55D    B: #D0E559    C: #CCE55E    D: #23242D    **Ans: A**

| Answer with Color Cue | Answer without Color Cue |
| --- | --- |
| A | B / D |

Figure 63: **Color clues play as a critical role for Color Extraction task**. Option backgrounds correspond to their color codes.

**Object Recognition**

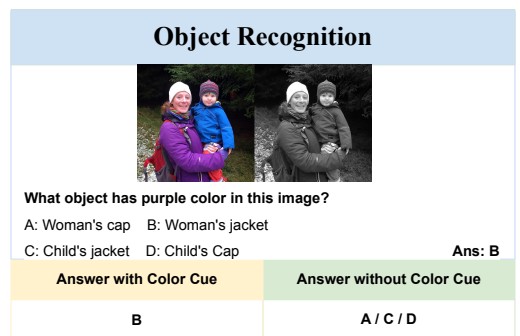

What object has purple color in this image?

A: Woman's cap    B: Woman's jacket
C: Child's jacket    D: Child's Cap    **Ans: B**

| Answer with Color Cue | Answer without Color Cue |
| --- | --- |
| B | A / C / D |

Figure 64: **Color clues play as a critical role for Object Recognition task**.

**Color Proportion**

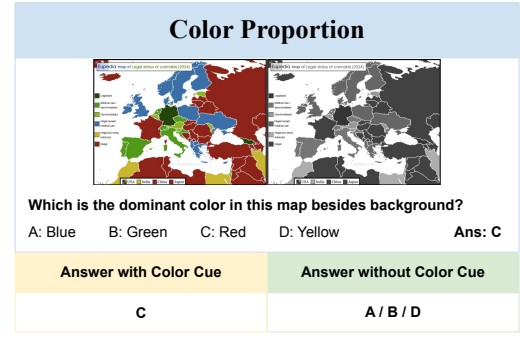

Which is the dominant color in this map besides background?

A: Blue    B: Green    C: Red    D: Yellow    **Ans: C**

| Answer with Color Cue | Answer without Color Cue |
| --- | --- |
| C | A / B / D |

Figure 65: **Color clues play as a critical role for Color Proportion task**.

**Color Comparison**

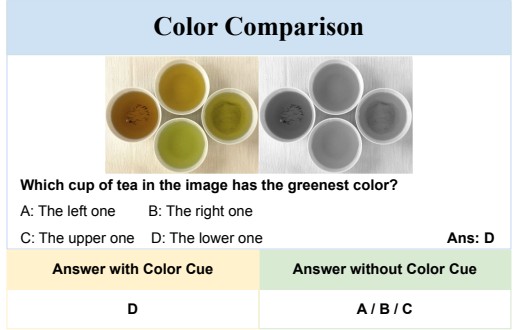

Which cup of tea in the image has the greenest color?

A: The left one    B: The right one
C: The upper one    D: The lower one    **Ans: D**

| Answer with Color Cue | Answer without Color Cue |
| --- | --- |
| D | A / B / C |

Figure 66: **Color clues play as a critical role for Color Comparison task**.

**Color Counting**

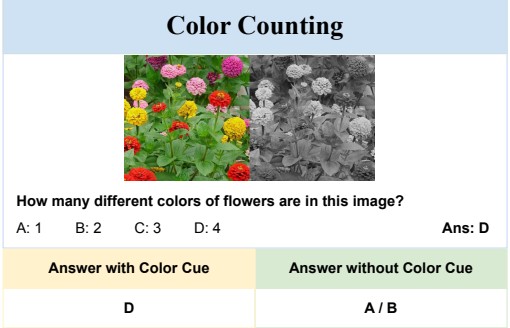

How many different colors of flowers are in this image?

A: 1    B: 2    C: 3    D: 4    **Ans: D**

| Answer with Color Cue | Answer without Color Cue |
| --- | --- |
| D | A / B |

Figure 67: **Color clues play as a critical role for Color Counting task**.

**Object Counting**

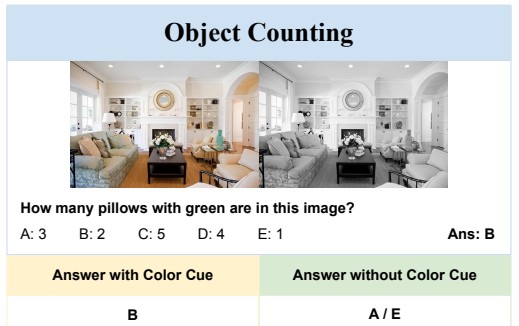

How many pillows with green are in this image?

A: 3  B: 2  C: 5  D: 4  E: 1    **Ans: B**

| Answer with Color Cue | Answer without Color Cue |
|---|---|
| B | A / E |

Figure 68: **Color clues play as a critical role for Object Counting task**.

**Color Blindness**

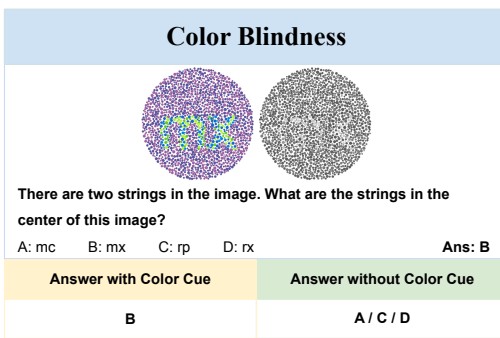

There are two strings in the image. What are the strings in the center of this image?

A: mc  B: mx  C: rp  D: rx    **Ans: B**

| Answer with Color Cue | Answer without Color Cue |
|---|---|
| B | A / C / D |

Figure 69: **Color clues play as a critical role for Color Blindness task**.

**Color Illusion**

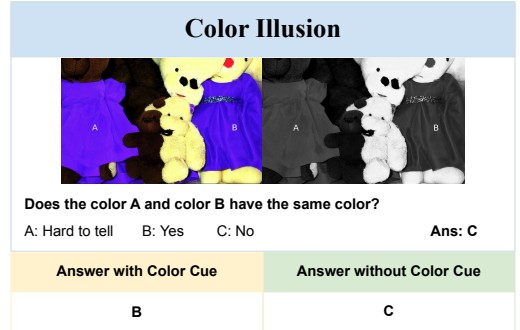

Does the color A and color B have the same color?

A: Hard to tell  B: Yes  C: No    **Ans: C**

| Answer with Color Cue | Answer without Color Cue |
|---|---|
| B | C |

Figure 70: **Color clues negatively affect VLMs prediction for Color Illusion task**.

**Color Mimicry**

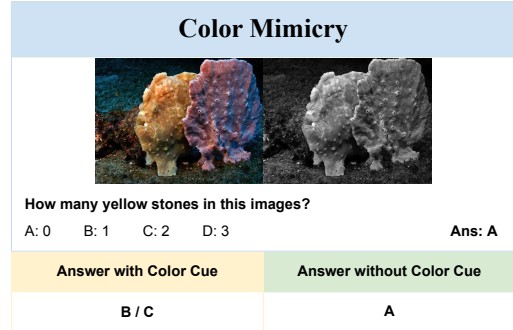

How many yellow stones in this images?

A: 0  B: 1  C: 2  D: 3    **Ans: A**

| Answer with Color Cue | Answer without Color Cue |
|---|---|
| B / C | A |

Figure 71: **Color clues negatively affect VLMs prediction for Color Mimicry task**.

## M.3 Failure with LLM and Vision

We present a representative failure case that highlights limitations in both the vision and language components of the model. As shown in Figure 72, the model fails to correctly interpret the visual content—it misidentifies the target colors by focusing on pink and purple flowers instead of red and yellow ones, indicating a vision encoder error. Furthermore, the language model compounds this mistake by generating an incorrect chain-of-thought reasoning and arriving at an erroneous answer based on the wrong color categories. This example underscores the necessity of evaluating both visual perception and language reasoning when diagnosing failure modes in vision-language models.

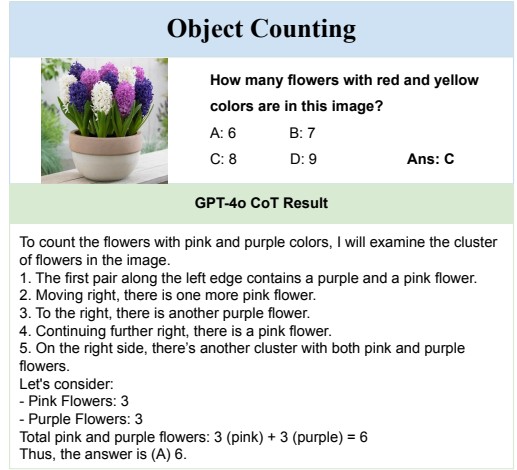

**Object Counting**

How many flowers with red and yellow colors are in this image?

A: 6  B: 7

C: 8  D: 9  **Ans: C**

**GPT-4o CoT Result**

To count the flowers with pink and purple colors, I will examine the cluster of flowers in the image.
1. The first pair along the left edge contains a purple and a pink flower.
2. Moving right, there is one more pink flower.
3. To the right, there is another purple flower.
4. Continuing further right, there is a pink flower.
5. On the right side, there's another cluster with both pink and purple flowers.
Let's consider:
- Pink Flowers: 3
- Purple Flowers: 3
Total pink and purple flowers: 3 (pink) + 3 (purple) = 6
Thus, the answer is (A) 6.

Figure 72: **Case that model fails because of both vision encoder and language model.**

## M.4  Easy Cases

We present samples cases that majority of VLMs reach the correct answers.

### Color Recognition

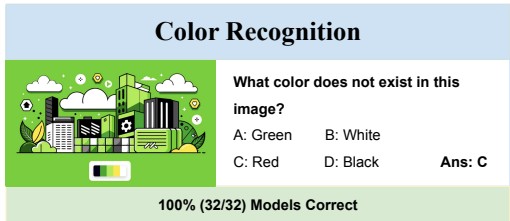

What color does not exist in this image?

A: Green    B: White

C: Red    D: Black    **Ans: C**

**100% (32/32) Models Correct**

Figure 73: **Color Recognition case that majority of VLMs provide correct results**.

### Color Extraction

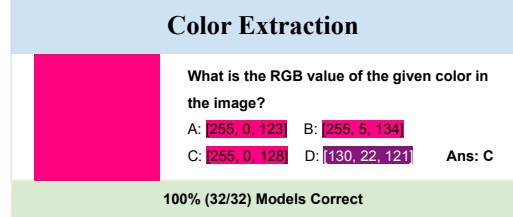

What is the RGB value of the given color in the image?

A: [255, 0, 123]    B: [255, 5, 134]

C: [255, 0, 128]    D: [130, 22, 121]    **Ans: C**

**100% (32/32) Models Correct**

Figure 74: **Color Extraction case that majority of VLMs provide correct results**. Option backgrounds correspond to their color codes.

### Object Recognition

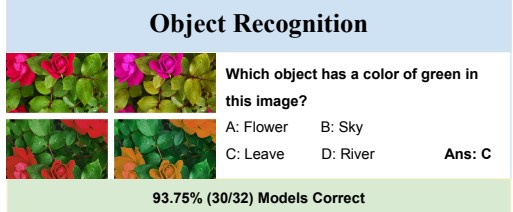

Which object has a color of green in this image?

A: Flower    B: Sky

C: Leave    D: River    **Ans: C**

**93.75% (30/32) Models Correct**

Figure 75: **Object Recognition case that majority of VLMs provide correct results**.

### Color Proportion

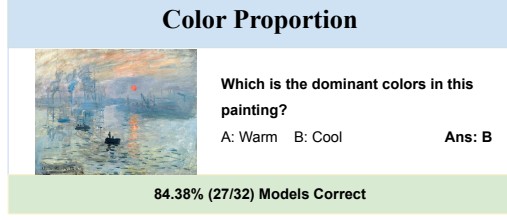

Which is the dominant colors in this painting?

A: Warm    B: Cool    **Ans: B**

**84.38% (27/32) Models Correct**

Figure 76: **Color Proportion case that majority of VLMs provide correct results**.

### Color Comparison

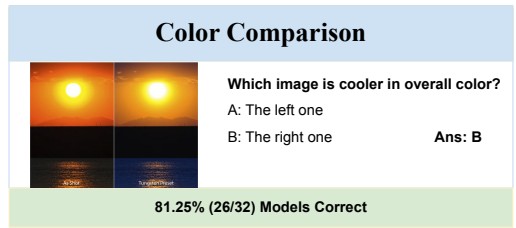

Which image is cooler in overall color?

A: The left one

B: The right one    **Ans: B**

**81.25% (26/32) Models Correct**

Figure 77: **Color Comparison case that majority of VLMs provide correct results**.

### Object Counting

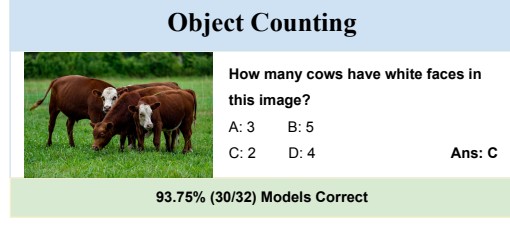

How many cows have white faces in this image?

A: 3    B: 5

C: 2    D: 4    **Ans: C**

**93.75% (30/32) Models Correct**

Figure 78: **Object Counting case that majority of VLMs provide correct results**.

### Color Mimicry

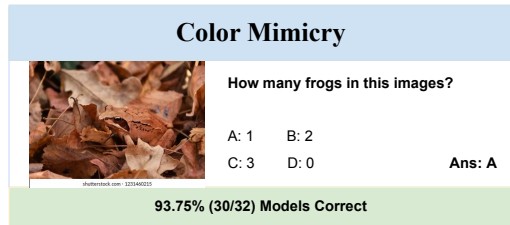

How many frogs in this images?

A: 1    B: 2

C: 3    D: 0    **Ans: A**

**93.75% (30/32) Models Correct**

Figure 79: **Color Mimicry case that majority of VLMs provide correct results**.

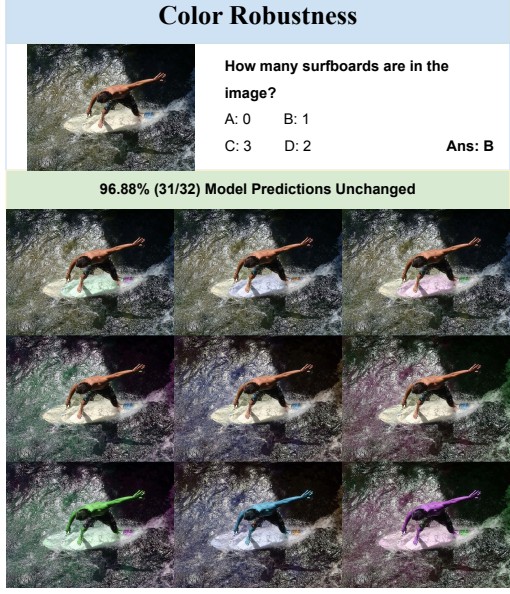

Figure 80: **Color Robustness case that majority of VLMs provide unchanged results over color variations in images**.

## M.5  Difficult Cases

We present samples cases that majority of VLMs reach the incorrect answers.

### Color Recognition

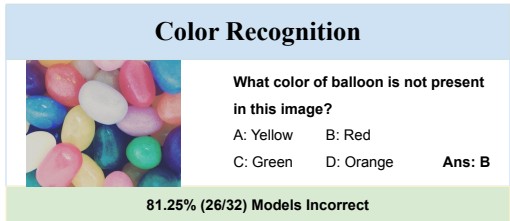

**What color of balloon is not present in this image?**

A: Yellow      B: Red

C: Green      D: Orange      **Ans: B**

**81.25% (26/32) Models Incorrect**

Figure 81: **Color Recognition case that majority of VLMs provide incorrect results**.

### Color Extraction

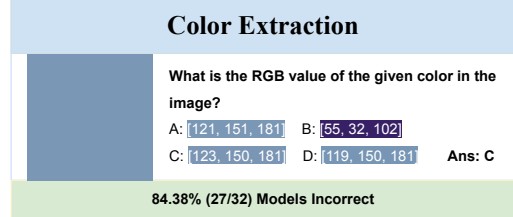

**What is the RGB value of the given color in the image?**

A: [121, 151, 181]   B: [55, 32, 102]

C: [123, 150, 181]   D: [119, 150, 181]   **Ans: C**

**84.38% (27/32) Models Incorrect**

Figure 82: **Color Extraction case that majority of VLMs provide incorrect results**. Option backgrounds correspond to their color codes.

### Object Recognition

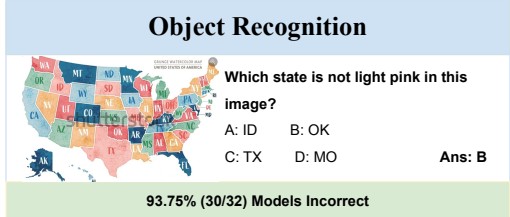

**Which state is not light pink in this image?**

A: ID      B: OK

C: TX      D: MO      **Ans: B**

**93.75% (30/32) Models Incorrect**

Figure 83: **Object Recognition case that majority of VLMs provide incorrect results**.

### Color Proportion

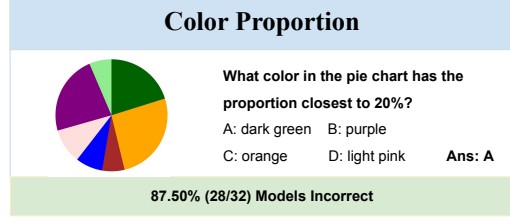

**What color in the pie chart has the proportion closest to 20%?**

A: dark green   B: purple

C: orange      D: light pink      **Ans: A**

**87.50% (28/32) Models Incorrect**

Figure 84: **Color Proportion case that majority of VLMs provide incorrect results**.

### Color Comparison

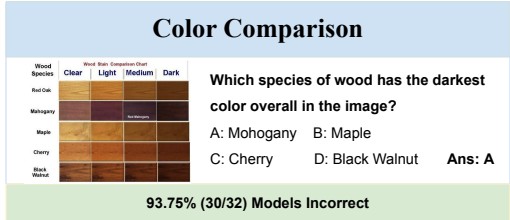

**Which species of wood has the darkest color overall in the image?**

A: Mohogany   B: Maple

C: Cherry      D: Black Walnut   **Ans: A**

**93.75% (30/32) Models Incorrect**

Figure 85: **Color Comparison case that majority of VLMs provide incorrect results**.

### Color Counting

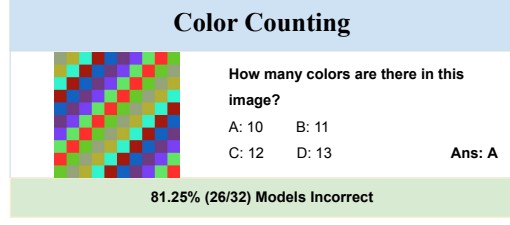

**How many colors are there in this image?**

A: 10      B: 11

C: 12      D: 13      **Ans: A**

**81.25% (26/32) Models Incorrect**

Figure 86: **Color Counting case that majority of VLMs provide incorrect results**.

### Object Counting

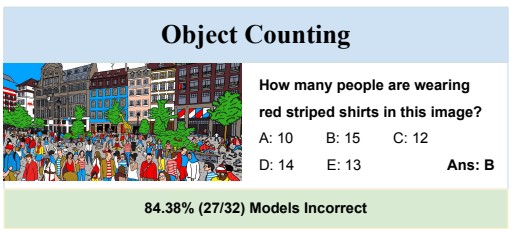

**How many people are wearing red striped shirts in this image?**

A: 10      B: 15      C: 12

D: 14      E: 13      **Ans: B**

**84.38% (27/32) Models Incorrect**

Figure 87: **Object Counting case that majority of VLMs provide incorrect results**.

### Color Illusion

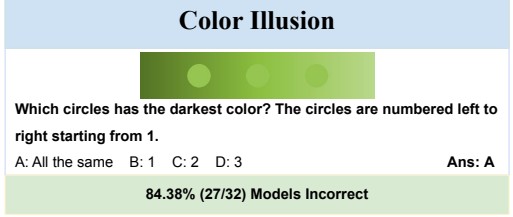

**Which circles has the darkest color? The circles are numbered left to right starting from 1.**

A: All the same   B: 1   C: 2   D: 3      **Ans: A**

**84.38% (27/32) Models Incorrect**

Figure 88: **Color Illusion case that majority of VLMs provide incorrect results**.

**Color Mimicry**

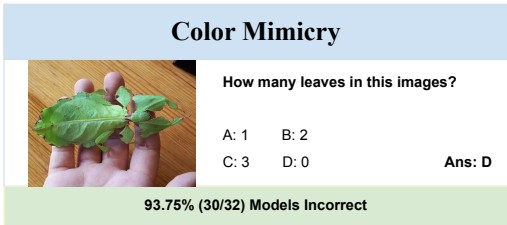

How many leaves in this images?

A: 1  B: 2

C: 3  D: 0      **Ans: D**

**93.75% (30/32) Models Incorrect**

Figure 89: **Color Mimicry case that majority of VLMs provide incorrect results**.

**Color Blindness**

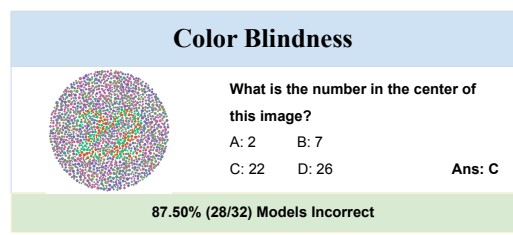

What is the number in the center of this image?

A: 2  B: 7

C: 22  D: 26    **Ans: C**

**87.50% (28/32) Models Incorrect**

Figure 90: **Color Blindness case that majority of VLMs provide incorrect results**.

**Color Robustness**

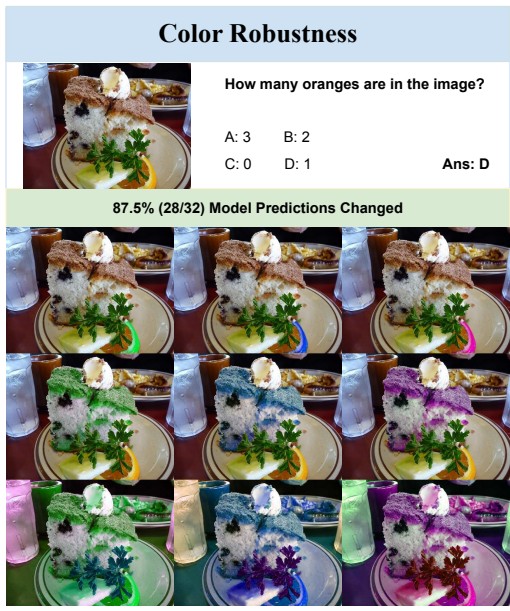

How many oranges are in the image?

A: 3  B: 2

C: 0  D: 1     **Ans: D**

**87.5% (28/32) Model Predictions Changed**

Figure 91: **Color Robustness case that majority of VLMs change the answers over color variations in images**.

