# OpenReview forum: "ColorBench: Can VLMs See and Understand the Colorful World? A Comprehensive Benchmark for Color Perception, Reasoning, and Robustness"
_NeurIPS.cc/2025/Datasets_and_Benchmarks_Track — NeurIPS 2025 Datasets and Benchmarks Track poster_

### Official Review · Reviewer_123a · 2025-07-02

**Rating:** 4
**Confidence:** 4

**Summary:**

This paper proposes COLORBENCH, which seems to be the first benchmark really focused on evaluating color understanding in vision-language models (VLMs). The benchmark is quite broad: it includes color perception tasks, reasoning tasks, and robustness tests where images are recolored. The authors evaluate 32 models, both open and proprietary, across 11 tasks and over 1,400 examples. They find that even the strongest models perform much worse than humans, and sometimes removing colors (grayscale) even helps the models, which is kind of surprising. They plan to release the benchmark and the code.

**Dataset Code Accessibility:**

Partly

**Ethical Considerations:**

No, there are no or only very minor ethics concerns

**Final Justification:**

After reviewing the rebuttal, engaging in discussions with the authors, and considering input from other reviewers and the AC, I have revised my recommendation from borderline reject to borderline accept.
1) The authors have expanded the discussion on why certain models fail and why scaling laws offer limited benefits, adding both quantitative evidence and qualitative examples that make the analysis more convincing.
2) They addressed concerns regarding the realism of synthetic recolored images by providing additional visual samples and explaining the steps taken to ensure naturalness, which alleviates my worry about major dataset biases.
3) Clarifications on the recoloring pipeline and task definitions, along with the inclusion of clearer illustrations, improved readability and made the methodology easier to follow.

**Limitations Weaknesses:**

1.	Some parts of the paper feel a bit rushed, like the analysis of why certain models fail or why scaling laws don’t help much. It could use more qualitative examples or deeper digging.
2.	The dataset mixing real images and synthetic recoloring might introduce biases, but the paper doesn’t discuss this much. For example, are the recolored images realistic or sometimes unnatural?
3.	It’s quite a big benchmark, and I wonder if smaller labs will be able to easily run all 11 tasks. Maybe some guidance on subsets would help.
4.	The novelty is mostly in the benchmark itself. The modeling or methodological contributions are limited.
5.	Some parts of the recoloring pipeline and task definitions are dense; it took me a couple readings to parse them. A clearer diagram or concrete examples would help.

**Strengths Contributions:**

1.	The paper addresses a clear gap—most benchmarks don’t look at color understanding in any systematic way, and in many real applications color is crucial.
2.	The benchmark design is quite comprehensive, probably more so than anything else out there on this topic, with multiple tasks spanning perception, reasoning, and robustness.
3.	The experiments are extensive—32 models is a lot to compare, and the results show convincingly that this is a challenging area.
4.	I like that the authors went beyond just perception to include illusions and color robustness, which is an underexplored angle.
5.	Making all this data public will be helpful for the community and probably spur follow-up work.

---

> ### Author Rebuttal · Authors · 2025-07-31
>
> We sincerely appreciate the time and effort the reviewer took to evaluate our manuscript and provide valuable feedback! We thank the recognition of benchmark’s comprehensiveness, and our efforts in broad task coverage, large-scale model evaluation, and exploration of underexamined aspects.
>
> > Q1: Some parts of the paper feel a bit rushed, like the analysis of why certain models fail or why scaling laws don’t help much. It could use more qualitative examples or deeper digging.
>
> We sincerely appreciate your kind advice, we will modify the analysis sections accordingly in our later version.
>
> Regarding **VLM failure cases**, we currently present integrated-level examples in Appendix J.4 and J.5, highlighting instances commonly answered correctly or incorrectly by most VLMs. To further deepen our analysis, we will introduce an additional discussion section featuring case analyses for representative models, thereby providing detailed model-level insights.
>
> Concerning the limited effectiveness of scaling laws, we address this point in Appendix at Line 271. To elaborate further, the relatively modest **impact of language model** scaling arises because color understanding predominantly relies on visual perception, mediated chiefly by the vision encoder. Although increasing language model size can enhance feature utilization, **the benefit of scaling is less pronounced for tasks centered on direct visual perception compared to tasks involving abstract reasoning.** This accounts for the subtle yet noticeable scaling trend in our results. Regarding the **vision encoder**, our analysis is constrained by the limited diversity of open-source models. **Many models share similar size or types of vision encoder, making it difficult to isolate the effect of vision architecture on color understanding.** Moreover, existing training datasets often do not explicitly emphasize color information, meaning that increasing model size alone does not necessarily lead to improved performance on color-specific tasks. We will incorporate this detailed discussion into the new version of our manuscript to enhance the clarity and rigor of our insights.
>
> Additionaly, as suggested by Reviewer Ybzm, we further conduct in-depth analysis of VLM failures using attention visualization techniques. Detailed outcomes from this analysis will be integrated into the upcoming version of our paper.
>
> > Q2: Bias introduced by mixing real imagesa and synthetic recolored images, and whether the recolored images appear realistic or sometimes unnatural.
>
> In ColorBench, we carefully design the use of synthetic recoloring to minimize potential biases. Specifically, the **Perception** and **Reasoning** categories include only the original real images to ensure visual realism. Synthetic recoloring is used exclusively in the **Robustness** category, where the goal is to evaluate model sensitivity to color variations. For this category, the un-realistic or unnatural property of these recolored images is actually desired, as we try to evaluate the robustness of models when facing this unnatural color changes.
>
> As detailed in Table 6, we apply three controlled recoloring strategies:
> - **Entire Image**: recolor the whole image,
> - **Target Segment**: randomly recolor one object,
> - **Largest Segment**: recolor the largest object.
>
> To create the color variations, we shift the hue by 90°, 180°, or 270° to cover a broad range of the color spectrum (see Line 584). Importantly, we only alter color attributes **without modifying object shapes or contextual elements**, preserving the overall realism of the image.
> Furthermore, our question and answer design ensures that **questions are color-invariant**, which means answers remain valid regardless of whether the recoloring appears fully natural. This helps isolate color variation as the variable of interest while minimizing confounding factors. We will make this discussion more explicit in the paper to clarify our design choices and mitigation of potential biases.
>
> > Q3: The current benchmark size is large for smaller labs, and some guidance on subsets would help.
>
> We note that the **Perception and Reasoning** categories contain approximately 100 samples per task, making them lightweight and accessible for smaller labs.
>
> The **Robustness** category includes 4.9k samples, which are systematically derived from the same base images with controlled variations across two axes:
> - **Recolored area**: entire image, randomly selected object, or largest object,
> - **Recolored hue shift**: 90°, 180°, and 270°.
>
> To support more flexible usage, we will provide guidance on recommended subsets. For instance, to evaluate robustness to global color shifts, users can test only on the original image and its corresponding entire-image recolored versions. Similarly, object-level robustness can be evaluated by selecting one type of object recoloring (e.g., largest segment). We will also include annotation indicating the recoloring strategy for each sample to support customized benchmarking.
>
> > Q4: The novelty is mostly in the benchmark itself. The modeling or methodological contributions are limited.
>
> As this submission is to the benchmark track, we mainly focus on the construction and evaluation of the benchmark itself. Except for this, we are the first to conduct **a comprehensive study of the lmitations of current VLM on color understanding**.
> We also contribute **a controlled data synthesis pipeline that enables fine-grained color variation**, allowing targeted evaluation of robustness. In the experiment section, we perform in-depth analysis of model behavior across perception, reasoning, and robustness tasks and provide multiple insights based on the evaluation results. For example, our comparison between VLMs and human annotators highlights a substantial performance gap, suggesting that color understanding is underrepresented in current training paradigms.
> We believe these insights and tools not only benchmark current models but also provide practical guidance for future VLM development that better incorporates visual attribute understanding.
>
> In addition, as suggested by Reviewer y1iS, we further conduct experiments evaluating how finetuning on color-related tasks improve the benchmark performance, and we will include this analysis in the discussion of our paper.
> We sincerely hope the reviewer can re-evaluate the contribution of our paper.
>
>  > Q5: Some parts of the recoloring pipeline and task definitions are dense; A clearer diagram or concrete examples would help.
>
> Thank you for the helpful suggestion.
> To support understanding, we currently provide **task definitions with example prompts** in Figure 1 and **concise descriptions** in Table 7. For the recoloring pipeline, Figure 3 outlines the process, and Figure 16 in the appendix shows **representative examples of recolored outputs**. A more **detailed explanation** is also included in Appendix C due to page limit for main context.
> To further improve clarity, we will revise and integrate Figure 3 and Figure 16 into a single, clearer diagram with step-by-step annotations and example outputs.

---

### Official Review · Reviewer_y1iS · 2025-07-02

**Rating:** 4
**Confidence:** 3

**Summary:**

This paper introduces COLORBENCH, a novel and comprehensive benchmark designed to evaluate the color understanding capabilities of Vision-Language Models (VLMs). The authors argue that while color is a fundamental aspect of human visual perception and reasoning, it has been largely overlooked in existing VLM evaluation suites.

**Dataset Code Accessibility:**

Yes

**Ethical Comments:**

There are no ethical concerns since the data and model are all publicly available.

**Ethical Considerations:**

No, there are no or only very minor ethics concerns

**Final Justification:**

I maintain the initial score since the majority of my concerns have been addressed.

**Limitations Weaknesses:**

Studies using MegaCOIN have shown that even advanced VLMs underperform in color recognition compared to human benchmarks, but fine-tuning on specialized datasets significantly improves their performance. So it may be more thoughtful to fine-tune some existing vlms and then see whether they can understand better for color perception and reasoning.

[1] MEGACOIN: Enhancing Medium-Grained Color Perception for Vision-Language Models (https://arxiv.org/pdf/2412.03927)

**Strengths Contributions:**

1. This work addresses a clear and important gap in VLM evaluation. As VLMs are deployed in more real-world applications (e.g., medical imaging, remote sensing, e-commerce), a nuanced understanding of color becomes critical. COLORBENCH provides a well-motivated and thoughtfully constructed tool for the community.

2. Comprehensive and Well-Structured Benchmark: The taxonomy of Perception, Reasoning, and Robustness is logical and provides a holistic view of the problem. The 11 sub-tasks are creative and cover a wide range of challenges, from basic recognition to complex reasoning (e.g., color blindness tests, illusions, mimicry) grounded in practical applications.

3. The paper is exceptionally well-written and easy to follow. The figures, particularly Figure 1, provide an excellent overview of the benchmark's scope and tasks. The appendices are detailed, providing the necessary information for reproducibility and further analysis.

---

> ### Author Rebuttal · Authors · 2025-07-31
>
> We sincerely appreciate the reviewer’s time, effort, and thoughtful feedback! We also thank reviewer for recognizing the importance of our benchmark. We appreciate the acknowledgment of our comprehensive task design, and the usefulness of our figures and appendices for reproducibility and practical application.
>
>
> > Q1: It may be more thoughtful to fine-tune existing VLMs with specialized datasets and then see whether model can understand better for color perception and reasoning.
>
> Thank you for your insightful comment. As this submission is to the benchmark track, our primary focus is on the benchmark itself and the insights that can be gained from comprehensive evaluation using it.
> While we agree that fine-tuning VLMs can enhance their performance on color-related tasks, the limited data size of ColorBench, particularly for tasks relying on human annotation, constrains the feasibility of effective fine-tuning. Specifically for Color Perception and Reasoning categories, our benchmark contains 10 tasks, each with around 100 samples, which is insufficient for fine-tuning on a per-task basis.
> To partially address this limitation, we conduct fine-tuning experiments on tasks with synthetic data in ColorBench, including Color Extraction, Color Illusion, and Color Blindness. Using our existing synthetic data generation pipeline used for ColorBench, we generate a training dataset for these tasks to support fine-tuning experiments.
>
> Table 2: **Number of synthetic samples generated** for fine-tuning experiments.
> | Task            | Num of Generated Data |
> |-----------------|------------------------|
> | Color Extraction | 2400                  |
> | Color Illusion   | 2400                  |
> | Color Blindness  | 2280                  |
>
> We experiment with Qwen2.5-VL-3B and Qwen2.5-VL-7B under the following settings:
> - **Vision encoder only**: Fine-tuning the vision encoder while keeping other components fixed.
> - **MLP only**: Fine-tuning only the MLP layers.
> - **LoRA**: Fine-tuning both the vision encoder and the language model using LoRA.
>
> The evaluation results with finetuned VLMs is shown below (in Table 3). These results indicate that LoRA-based fine-tuning yields the best performance, with a significant increase in overall accuracy.
> Notably, fine-tuning on these three tasks significantly enhanced model accuracy, not only on targeted tasks (Color Extraction, Color Illusion, Color Blindness) but also positively influenced related tasks such as Color Proportion, Color Comparison, and Color Counting.
> However, the funetuning process has a limited or negative effects on some tasks, suggesting that fine-tuning exclusively on these color-focused tasks might not universally enhance all capabilities necessary for comprehensive color perception and reasoning.
>
> Table 3: **Accuracy (%) of Qwen2.5-VL models (3B and 7B) under different fine-tuning strategies** across ColorBench tasks. Bold values indicate the highest score within each model group across the three fine-tuning settings.
>
> | Model           | Finetune Settings | Color Recognition | Color Extraction | Object Recognition | Color Proportion | Color Comparison | Color Counting | Object Counting | Color Illusion | Color Mimicry | Color Blindness | Perception Acc | Reasoning Acc | Overall Acc |
> |----------------|-------------------|--------------------|-------------------|---------------------|------------------|-------------------|----------------|------------------|----------------|----------------|------------------|-----------------|----------------|--------------|
> | Qwen2.5-VL-3B   | No Finetune       | 72.37             | 38.54            | 74.03              | 43.75           | 48.51            | 22.55         | 25.24           | 43.01         | 45.71         | 24.20           | 59.84      | **34.42**     | 41.05       |
> |                | Vision encoder     | **75.00**         | 43.75            | **76.62**          | **47.50**       | 51.49            | 21.36         | 27.18           | **46.24**         | 47.14         | 26.75           | 57.11          | 31.77         | 43.51   |
> |                | MLP                | **75.00**         | 43.75            | 74.03              | 46.25           | **53.47**        | 23.30         | 27.18           | 45.16         | 47.14         | 26.11           | 56.88          | 31.77         | 43.41       |
> |                | LoRA               | 72.37             | **75.00**        | 71.43              | 45.00           | 51.49            | **24.27**     | **32.04**       | **46.24**         | **50.00**     | **28.03**       | **62.53**      | 33.72     | **47.07**   |
> | Qwen2.5-VL-7B   | No Finetune       | 76.32             | 48.96            | **84.42**          | 47.50           | 52.48            | 19.61         | **33.98**       | 44.09         | **55.71**         | 28.66           | 68.27      | **38.39**     | 46.18       |
> |                | Vision encoder     | **80.26**         | 50.00            | 83.12              | 48.75           | **56.44**        | 20.39         | **33.98**       | 50.54         | **55.71**         | 26.11           | 62.53          | 34.11         | 47.28       |
> |                | MLP                | 75.00             | 43.75            | 74.03              | 46.25           | 53.47            | **23.30**     | 27.18           | 45.16         | 47.14         | 26.11           | 56.88          | 31.77         | 43.41       |
> |                | LoRA               | 77.63             | **82.29**        | 83.12              | **50.00**       | 55.45            | **23.30**     | 31.07           | **52.69**     | **55.71**     | **33.12**       | **69.30**      | 36.45     | **51.67**   |
>
> Due to the limited rebuttal time, we will provide a more comprehensive version of experiments and analysis in the discussion of the paper.

---

> > ### Comment · Reviewer_y1iS · 2025-08-07
> >
> > Thanks for the detailed clarification. Most of my concerns are addressed.

---

### Official Review · Reviewer_Ybzm · 2025-07-02

**Rating:** 5
**Confidence:** 3

**Summary:**

This paper studies the colour understanding ability of existing VLMs, ranging from lightweight models to large proprietary models. It proposes a benchmark consisting of three main tasks, which are further divided into eleven subtasks. Each subtask evaluates whether VLMs can perceive colour similarly to humans from diverse perspectives.

**Dataset Code Accessibility:**

Yes

**Dataset Code Comments:**

Available on Hugging Face.

**Ethical Considerations:**

No, there are no or only very minor ethics concerns

**Final Justification:**

Before the rebuttal, my major concerns are the limited discussion of potential factors that affect colour understanding and multimodality contributions. The authors have addressed my concerns. I maintain my score.

**Limitations Weaknesses:**

1. Although this paper presents thorough results, its analysis is limited, focusing only on model size. It would be valuable to discuss other potential factors that might influence colour understanding.

2. The discussion of results is restricted to quantitative analysis. A deeper investigation is expected to understand why VLMs make mistakes and why colour may negatively influence judgment in certain scenarios — for example, by analysing attention maps.

3. VLMs access information from both vision and text. Therefore, it would be helpful to see results that isolate the effects of each modality. It appears that colour information is primarily extracted by the vision module, while the final predictions are mainly produced by the language module. The contribution of each model is unclear in making the final answer.

**Strengths Contributions:**

1. A comprehensive benchmark for evaluating the colour understanding ability of VLMs is established. Moreover, this benchmark is reasonable and well-motivated. Its results reveal that there is still a noticeable gap between VLMs and human vision, which provides potential research targets for VLMs.

2. The proposed benchmark is well-presented, and the writing throughout the paper is concise.

3. Beyond reporting results, the paper additionally investigates the effects of model size and chain-of-thought (CoT) reasoning on colour understanding.

4. The dataset construction process and dataset samples are described in detail.

---

> ### Author Rebuttal · Authors · 2025-07-31
>
> We appreciate that reviewer's encouraging review and finds our benchmark well-motivated, and our analysis of model scaling and reasoning effects insightful. We’re also grateful for the recognition of the detailed dataset construction and clear presentation.
>
> > Q1: Discussion of potential factors that might influence colour understanding
>
> We agree that further analysis including more factors could enrich the discussion.
> Beyond model size, our current paper also analyzes other factors related to color understanding. Specifically, Finding 3 investigates the **effect of adding reasoning process** toward color understanding, while Finding 4 explores **the impact of color itself** through comparative experiments on grayscale and colorful images.
>
> Regarding model-related factors (e.g., architecture or vision encoder type), the current available open-source VLMs provide **limited diversity in architecture**, as most models use similar vision encoders and differ primarily in language model size. This constrains us to systematically explore the effect of other architectural factors. As more diverse VLMs become available, we will try to conduct a broader and deeper discussion related to model architectural factors in future work.
>
> > Q2: The discussion of results is restricted to quantitative analysis. Expecting a deeper investigation to understand why VLMs make mistakes and why colour may negatively influence judgment in certain scenarios, i.e. by analysing attention maps.
>
> In our current submission, we provide **qualitative insights through case studies in Appendix J**, including a failure analysis incorporating reasoning steps (Line 687), as well as specific examples illustrating errors originating from vision or language components of VLMs (Line 706).
>
> Furthermore, we recognize the potential of attention map analysis for interpreting model decisions.
> To gain deeper insights into VLM failures on color-related tasks, we perform a case analysis using **Qwen2.5-VL-3B and 7B models** on Color Recognition and Object Recognition tasks, following the **attention visualization methodology** introduced in [2]. We specifically select examples where the 3B model failed but the 7B model succeeded, allowing us to better isolate and understand the performance discrepancies.
> Our preliminary findings shows that in only a few failure cases, the models fail due to incorrect object localization. **In most cases, VLMs correctly attend to the relevant regions but still produce incorrect answers.** This might indicate that VLMs’ cannot accurately interpret color information, rather than deficiencies in visual grounding. We will try the visual cropping methods in [2] to further investigate the failure reason.
> Due to the very limited time for rebuttal, it's hard for us conduct through detailed experiments on VLM attention maps. We will explore this approach in future work to better understand the role of color in both correct and incorrect predictions.
>
> [2] Mllms know where to look: Training-free perception of small visual details with multimodal llms (https://arxiv.org/pdf/2502.17422)
>
>
> > Q3: The effects and contribution of each modality to the final answer.
>
> Thank you for raising this point regarding the importance of disentangling the contributions of vision and language modalities.
> In Finding 4, we investigate the **impact of color information within the visual modality** by comparing model performance on RGB versus **grayscale images**. This isolates the effect of color within the image modality.
> To further explore the contribution of the image modality, we conduct additional experiments using **only textual input** (i.e., questions and answer choices) by substituting the input images with pure black images of identical dimensions. This experiment involves three representative models (Qwen2.5-VL-7B, InternVL2.5-8B, and LLaVA-OV-7B) to analyze the effect of image toward final performance.
>
> Table 1: **Averaged Accuracy (%)** across three input settings (Text only, Grayscale image + text, RGB image + text) on Color Perception and Reasoning tasks.
> | Experiment Setting     | Color Recognition | Color Extraction | Object Recognition | Color Proportion | Color Comparison | Color Counting | Object Counting | Color Illusion | Color Mimicry | Color Blindness | Perception Acc | Reasoning Acc | Overall Acc |
> |------------------------|-------------------|------------------|--------------------|------------------|------------------|----------------|------------------|----------------|----------------|------------------|----------------|----------------|--------------|
> | Text only              | 27.63            | 47.57           | 38.53             | 24.17           | 34.65           | 21.04         | 21.04           | 41.58         | 37.14         | 26.96           | 38.37         | 25.54         | 31.49       |
> | Gray image + text      | 23.25            | 42.71           | 52.38             | 25.83           | 43.89           | 19.93         | 27.83           | 51.25         | 61.43         | 29.09           | 39.76         | 35.46         | 36.58       |
> | RGB image + text       | 75.00            | 50.00           | 83.12             | 50.00           | 56.11           | 21.57         | 31.07           | 42.29         | 52.38         | 23.99           | 67.87         | 37.44         | 45.38       |
>
>
> These results demonstrate that removing the visual modality (text-only setting) leads to the lowest performance across the majority of tasks in all three experiment settings.
> The performance difference among these three experiments helps to examine and disentangle the impact of textual input, image context (excluding color), and color information itself.
> Notably, in some tasks (Color Recognition, Object Recognition, and Color Proportion), the performance gap between text-only experiment and graysalce experiments is relatively small, whereas both are significantly outperformed by RGB image experiments. This suggests that **color cues play a substantially more important role than contextual visual or textual information** in these tasks.
> Due to the limited rebuttal time, the detailed version of experiments and analysis will be included in the new version of the paper.

---

> > ### Comment · Reviewer_Ybzm · 2025-08-03
> >
> > Dear Authors,
> >
> > Thank you for the detailed response. All of my concerns have been addressed, and I therefore maintain my score. I currently have no further concerns. I believe this is an interesting paper that investigates an important aspect of human vision (colour), and it holds significant value for the development and understanding of VLMs.

---

> > > ### Author Response · Authors · 2025-08-03
> > >
> > > Thank you for your reply and for recognizing the significance of our work. We greatly appreciate your time throughout the review process!

---

### Official Review · Reviewer_R3DC · 2025-07-05

**Rating:** 5
**Confidence:** 3

**Summary:**

ColorBench presents the first benchmark on evaluating VLMs on color understanding capabilities. They have bench-marked 32 VLMs on 11 / 12 tasks (Perception, Reasoning, and Robustness).

**Dataset Code Accessibility:**

Yes

**Dataset Code Comments:**

I could see the code in the git repo but hard to locate the dataset itself

**Ethical Comments:**

- reminded me of the story of Kokak films in mid mid-20th century were using reference cards for calibration; these cards were featured with white women with light skin. As a result darker skin tones were often underexposed, details in shadow were lost.

**Ethical Considerations:**

No, there are no or only very minor ethics concerns

**Limitations Weaknesses:**

Data Size; Of course ~5800 images should be enough to run a stable benchmark, e.g. for color robustness; however, it is always nice to have more ! perhaps a discussion on the limitation for data could be relevant.

**Strengths Contributions:**

- Good case to evaluate VLMs and they have tried different prompting techniques to evaluate the result.
- Good explanation on the case and a through discussion on the result.

---

> ### Author Rebuttal · Authors · 2025-07-31
>
> We sincerely appreciate the reviewer’s thoughtful feedback and acknowledges our diverse prompting strategies, good cases explanation and in-depth discussion of results!
>
> > Q1: Data size and discussion of the limitation of data
>
> In ColorBench, we collect, annotate and assess data manually for most domains, which ensures the quality of our benchmark but inevitably limits the overall dataset size.
> One potential method to address this in the future is to expand ColorBench through an automated data collection pipeline. However, since most of the current models do not perform well on the color-related tasks, how to construct trustworthy automatic data collection pipeline still needs further exploration.
> We will include a discussion on current size, style diversity, and category coverage of ColorBench in the paper's limitation part.
>
> > Q2: Hard to locate the dataset in git repo
>
> Thank you for pointing this out.
> Currently, the dataset download link is avalibale under the **“Download Dataset”** section of the GitHub repository. To further improve accessibility, we will add the dataset link to the top of the page and make it more prominent in the README page.

---

### Decision · Program_Chairs · 2025-09-18

**Decision:**

Accept (poster)

**Comment:**

The paper presents a technically sound dataset on a relatively underlooked problem of color understanding. The dataset is large and diverse, and no major issues are observed. All reviewers concur and the authors have addressed the minor issues raised by the reviewers adequately. I recommend a poster acceptance.